# Coherence of Global Hydroclimate Classification Systems

Kathryn L. McCurley Pisarello[1], James W. Jawitz[2]

[1]USDA-ARS, Southeast Watershed Research Laboratory, 2316 Rainwater Road, Tifton, Georgia 31793, USA
[2]Soil and Water Sciences Department, University of Florida, Gainesville, Florida 32611, USA

*Correspondence to*: Kathryn L. M. Pisarello (katie.pisarello@usda.gov)

**Abstract.** Climate classification systems are useful for investigating future climate scenarios, water availability, and even socioeconomic indicators as they relate to climate dynamics. There are several classification systems that apply water and energy variables to create zone boundaries, although there has yet to be a simultaneous comparison of the structure and function of multiple existing climate classification schemes. Moreover, there are presently no classification frameworks that include

evapotranspiration (ET) rates as a governing principle. Here, we developed a new system based on precipitation and potential evapotranspiration rates, as well as three systems based on ET rates, which were all compared against four previously established climate classification systems. The within-zone similarity, or coherence, of several long-term hydroclimate variables was evaluated for each system based on the premise that the interpretation and application of a classification framework should correspond to the variables that are most coherent. Additionally, the shape complexity of zone boundaries

was assessed for each system, assuming zone boundaries should be drawn efficiently such that shape simplicity and hydroclimate coherence are balanced for meaningful boundary implementation. The most frequently used climate classification system, Köppen-Geiger, generally had high hydroclimate coherence but also had high shape complexity. When compared to the Köppen-Geiger framework, the Water-Energy Clustering classification system introduced here showed overall improved or equivalent coherence for hydroclimate variables, yielded lower spatial complexity, and required only two,

compared to 24, parameters for its construction.

## 1 Introduction

A variety of classification schemes have been introduced to categorize specific biophysical characteristics of Earth systems, including those based on climatic behavior (Beck et al., 2018; Berghuijs and Woods, 2016; Holdridge, 1967), biodiversity (Olson et al., 2001), plant-climate interactions (Papagiannopoulou et al., 2018), and plant hardiness (Magarey et

al., 2008; McKenney et al., 2007). These frameworks classify elements of a system based on common atmospheric or terrestrial characteristics to maximize their within-zone similarity, or coherence, which allows for a transfer of understanding across zones of similar attributes (Lanfredi et al., 2019). This study focuses specifically on climate classification schemes, which have provided a climatic context for a variety of applications, including socioeconomic assessments of human health conditions (Boland et al., 2017; Jagai et al., 2007; Lloyd et al., 2007), economic development (Mellinger et al., 2000; Richards et al.,

2019), and the evaluation of anticipated terrestrial and climatic changes (Chen and Chen, 2013; Tapiador et al., 2019).

Different climate classification systems have emerged based on framework-specific suites of hydroclimatic variables used to define zone boundaries. Therefore, users should consider how potential classification system application corresponds to the variables used to create it (Knoben et al., 2018; Meybeck et al., 2013). Climate classification systems are usually based in part on annual and seasonal water-energy budgets (Beck et al., 2018; Berghuijs and Woods, 2016; Holdridge, 1967; Knoben et al., 2018; Meybeck et al., 2013). The Köppen-Geiger classification system, the most widely used climate framework, was developed to regionalize climatic variables (specifically accounting for seasonal precipitation and temperature) and is often employed to compare the output of global climate models (Peel et al., 2007; Tapiador et al., 2019). Another common system is the Holdridge Life Zones scheme, which was created to classify land area with respect to vegetation and soil (Holdridge, 1967). This system subdivides zones based on thresholds of annual precipitation (P), potential evapotranspiration (PET), biotemperature (growing season length and temperature), and latitude and altitude.

Recent work has extended climate classification frameworks to specifically encompass hydrologic factors, since water resources-based analyses should take place within relevant hydrologic boundaries (Knoben et al., 2018; Meybeck et al., 2013). For example, Meybeck et al. (2013) proposed a global zoning system that was primarily based on the mean temperatures and gauged runoff (Q) of river basins. They compared the resulting boundaries against the Köppen-Geiger and Holdridge frameworks to assess zone boundary overlaps. The authors also evaluated the within-zone coherence of mean annual temperature, P, and Q, concluding that the latter two were most coherent in dry zones and least coherent in equatorial zones, while temperature was most coherent in equatorial zones. However, Meybeck et al. (2013) did not compare their zone coherence to that of previously established systems. Similarly, Knoben et al. (2018) formed zone boundaries based on climate indices (average aridity, seasonality of aridity, and P as snow) with the objective of minimizing within-zone Q variability (i.e., maximizing Q coherence). Those authors compared their results to the Köppen-Geiger framework and found theirs to be more coherent with respect to flow regime, but they did not compare other water budget components nor additional climate classification systems.

Although the P and Q components of the long-term water budget have been extensively considered in climate classification schemes (Beck et al., 2018; Berghuijs and Woods, 2016; Holdridge, 1967; Knoben et al., 2018; Meybeck et al., 2013), notably absent is a system that is directly based on actual evapotranspiration (ET) rates. This gap is likely because ET traditionally has been the least empirically identified element of regional to global water budgets (Zhang et al., 2016). In addition to the absence of a zoning system that accounts for ET dynamics, there has been no comparison of within-zone hydroclimate coherence across multiple climate classification systems, with evaluation particularly lacking in considering ET rates. Furthermore, the spatial complexity of climate classification systems has not been systematically examined across multiple frameworks, although Guan et al., 2020 quantified the changing spatial structure of the Köppen-Geiger framework over time. Assessing the structure of a biophysical system is a concept that most notably originates from landscape ecology (O'Neill et al., 1988) and provides a suite of shape metrics that can be cross-disciplinarily applied. Quantifying shape pattern and spatial contouring of climate classification systems is important for understanding interactions between governing

hydroclimatic characteristics as well as anticipating socioecological consequences that result from changing atmospheric configurations (Guan et al., 2020).

This work seeks to provide empirical support for application-dependent selection among candidate climate classification systems. We suggest that a successful classification system should have high within-zone coherence for variables that are related to the system's intended use, combined with relatively low shape complexity across zones, which is best for ease of interpretation within management and policy contexts. As such, we postulate that for a given climate classification system, within-zone hydrologic coherence and inter-zone shape complexity will be closely related to the organizing principle of that system. For example, the Köppen-Geiger and Meybeck et al. (2013) systems are based in large part on P and Q, respectively, and therefore these systems should show high coherence for these variables. Similarly, zone shape complexity will be lower in classification systems that include spatial contiguity in the organizing criteria (e.g., Meybeck et al., 2013). Given the major gap regarding the inclusion of ET in climate classification systems, we also created a series of ET-based global classifications that were expected to yield comparatively higher ET coherence than other systems.

We evaluated within-zone coherence of long-term water budget components (mean annual ET, P, and Q) and synchronous P and PET seasonality, as well as zone shape complexity for four new global classification systems and compared these against four previously established systems (Beck et al., 2018; Holdridge, 1967; Knoben et al., 2018; Meybeck et al., 2013). The primary zone shape complexity metrics were the distribution of zone area (km$^2$), mean zone fragmentation (i.e., mean number of patches comprising each zone), and the number of zones required to effectively form hydroclimate boundaries. This work presents novel approaches to identify boundary complexities and determine appropriate applications of classification frameworks. Understanding the relevance of a climate classification system is important since such frameworks are used in multi-disciplinary contexts to examine hydrological, ecological, and societal phenomena.

## 2 Methods

### 2.1 Coherence and complexity metrics

Variable coherence is defined by within-zone variability, represented by the intra-zone coefficient of variation (CV) of the hydroclimate variable of interest. Lower CV values correspond to higher coherence, meaning that regions delineated by zone boundaries that yield low CV values are more spatially homogenous with respect to hydroclimate variables and are therefore more hydroclimatically continuous. An additional important component of this analysis is the evaluation of the tradeoffs between hydroclimate coherence and the shape complexity of zone boundaries. It is valuable to consider the structural attributes of zone boundaries because these boundaries are expected to change over time (Beck et al., 2018; Knoben et al., 2018). Building more precise boundaries may better delineate similar hydroclimate processes, but overly exact geographic specificity may compromise ease of interpretation, communication, and relevant application for management purposes (Knoben et al., 2018).

Classification system complexity metrics were primarily based on three principles: 1) classification systems should consist of a relatively even area distribution across zones, avoiding disproportionately large or small zones, 2) zones should be as hydrologically continuous as possible (Meybeck et al., 2013), minimizing patchiness or fragmentation, and 3) classification systems should comprise less than or equal to the number of zones in the Köppen-Geiger framework, which is used here as the standard to which other systems are compared. Therefore, complexity was assessed based on the inter-zone

distribution of area ($km^2$) as defined by CV, the mean number of patches in each zone (zone fragmentation), and the number of needed zones to bound hydroclimatically similar areas. The mean number of patches per zone was determined using the R function lsm_c_np in package *landscapemetrics* (Hesselbarth, et al., 2019). For each hydroclimate and complexity variable, statistical differences between classification systems were determined based on a series of two-sided Kolmogorov-Smirnov (K-S) tests, which compares probability distributions to a reference distribution.

**2.2 Database construction**

     Several open access datasets were compiled to create the database used for climate classification system calibration and validation. We evaluated global gridded monthly P and PET and mean annual ET and Q between 1980 and 2014 at a 0.5° x 0.5° spatial resolution. The Climate Research Unit TimeSeries V4.04 supplied monthly P and PET (Harris et al., 2020), while mean annual ET and Q were constructed from aggregated TerraClimate monthly data (Abatzoglou et al., 2018) by

summing long-term mean monthly values. In this case, long-term mean values muted interannual variability. Annual ET and Q were resampled from their original 1/24° x 1/24° resolution to the 0.5° x 0.5° resolution of P and PET.

     Additional ET and Q datasets were used for independent validation purposes. Observation-based monthly Q from 1980-2014 were obtained at 0.5° x 0.5° resolution from monthly global gridded runoff data (GRUN, Ghiggi et al., 2019). The Global Lobal Evaporation Amsterdam Model (GLEAM) produced terrestrial daily ET for 1980-2020 at 0.25° x 0.25°

resolution, which was also resampled to 0.5° x 0.5° resolution. Here, we used the updated GLEAM version 3.5a, which is based on ERA5 net radiation (satellite) and air temperature (reanalysis) datasets, downloadable at a monthly timestep (Martens et al., 2017). The GLEAM ET and GRUN Q datasets were independent from TerraClimate ET and Q datasets both temporally (Figures S1 and S3) and spatially (Figures S2 and S4). The two ET datasets were more similar than the two Q datasets, based on monthly linear models ($R^2$ ranging from 0.78 to 0.87 for ET and 0.47 and 0.84 for Q), and both ET and Q datasets showed

spatially consistent seasonal differences. Hereafter, TerraClimate ET and Q are simply referred to as "ET" and "Q" unless otherwise noted.

     Spatial analysis R packages *raster* (Hijmans, 2017)*, sp* (Bivand et al., 2013) and *ncdf4* (Pierce, 2017) were used to build the database of long-term monthly and annual averages. The spatial extent of this study comprised all global land areas, excluding Antarctica, which resulted in a total of 60,726 pixels.

**2.3 Sinusoidal functions as descriptors of seasonality**

The seasonal dynamics of monthly P and PET were additionally considered in this analysis, as they are also included in the Köppen-Geiger framework, which considers temperature as a general proxy for PET (Beck et al., 2018). Sine functions and their corresponding parameters can be used to describe intra-annual climate behavior. Sine functions were fitted to the long-term monthly distribution (following Berghuijs and Woods, 2016)

$$y_m = \bar{y}\left[1 + r_y \sin\left(\frac{2\pi(m-t_y)}{12}\right)\right] \tag{1}$$

where $y$ is either P or PET (mm month$^{-1}$) for each month, $m$, with overall monthly mean (i.e., mean of the 12 long term monthly means) denoted by the overbar, $r$ is dimensionless amplitude, and $t$ is the phase offset (months) from the reference time, January ($m = 1$). Phase difference, $\Delta t$, measures the synchronization of P and PET throughout the year, is determined as the difference $t_{PET} - t_P$, and is constrained $-6 \leq \Delta t \leq 6$ (more detail in SI).

Figure 1A shows the overall global distribution of $\Delta t$ (Equation S1), where some banding around the Tropics as well as the Middle East can be seen. Equation 1 yielded overall good fits to the long-term mean monthly distributions of P and PET, with R$^2$=0.67±0.28 and 0.84±0.18, respectively (mean±standard deviation across all pixels). These sine fits to monthly PET were statistically significant (p-value $\leqslant$ 0.05) in 97% of pixels, while fits to monthly P were statistically significant in 85% of pixels (Figure 1B-C). Cumulative distribution functions of both R$^2$ and p-value for PET and P sine fits can be seen in Figure S5. Similar to Berghuijs and Woods (2016), P fits were good in South America, and not as good in parts of the Sahara (Figure 1C). Our P fits were good in East Asia and not as good in the southern United States, while Berghuijs and Woods (2016) had more error in East Asia and less error in the United States. Lastly, compared to the performance of our PET fits shown in Figure 1B, the temperature fits of Berghuijs and Woods (2016) were overall much less spatially homogeneous than ours.

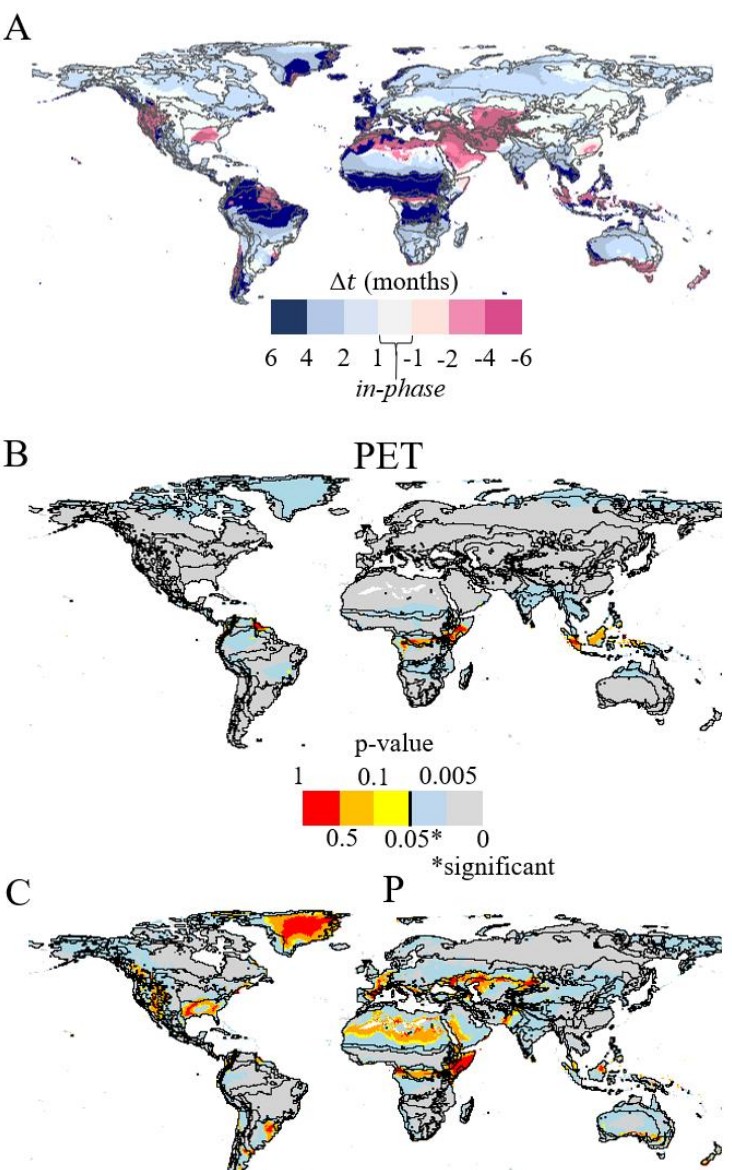

**Figure 1: Global spatial distributions of Δt (A) and of performance of monthly P (B) and PET (C) sine fits represented by p-value.**

## 2.4 Established climate classification systems

  Four previously established climate classification schemes were assessed in this analysis. We included two legacy schemes, Köppen-Geiger (KPG, Beck et al., 2018) and Holdridge Life (HDL, Holdridge, 1967) zoning systems, and two recently proposed frameworks, here referred to as Meybeck Hydroregion (MHR, Meybeck et al., 2013) and Knoben Hydroclimate (KHC, Knoben et al., 2018) systems. Note that the original KHC zones created by Knoben et al. (2018) were not delineated by discrete boundaries but were instead represented as pixels with a corresponding probability continuum of

belonging to a zone. However, Knoben et al., 2018 chose to bound 18 zones using their provided climate indices (aridity index,

seasonal aridity index, and precipitation as snow) for inter-system comparison purposes. In the present study, 18 KHC

boundaries were re-created using those climate indices in a clustering approach similar to the clustering methodology of

Knoben et al., 2018. Here we applied a k-means, multi-start clustering method (n=80 starts), which was also used to form

boundaries in two of our proposed frameworks described below. This k-means clustering approach, based on the Hartigan and

Wong (1979) algorithm, was employed using the kmeans function in the R package *stats* (R Core Team, 2018). Note that the

very small KPG zones "Csc" and "Cwc" did not appear in the 0.5° x 0.5° resolution KPG output created by Beck et al. (2018)

that was used in this study, resulting in 28 KPG presently analyzed zones. As in other climate classification studies (Knoben

et al., 2018; Meybeck et al., 2013), KPG was considered here to be the standard to which other systems are primarily compared

and evaluated for performance.

**2.5 Novel univariate ET climate classification systems**

This study establishes and verifies ET-relevant climate classification frameworks by creating zones primarily based

on ET rates and comparing ET coherence between systems. Three of the four systems developed in this study were univariate

(formatted from global mean annual ET rates) with a single condition to emphasize a specific optimization goal. A fourth

multivariate system is described below.

The first two novel univariate classification systems were based on the empirical cumulative distribution function

(CDF) for global long-term mean annual ET rates. The first classification system, ET Area-optimizing (ETA), was created

with the condition of having nearly equal area in each ET-based zone. This was motivated by the first complexity principle

described in Section 2.1, which states zones should not be meaninglessly small nor disproportionately large. The KPG system

has relatively high spatial non-uniformity, resulting in highly variable relevance for regional analyses. A classification system

that is more spatially uniform can better inform large spatial scale understanding as well as the application of regional to semi-

continental management strategies. Additionally, it is useful to have a simple baseline framework upon which to compare the

other ET-based systems. Ultimately, ETA is a system that seeks to maximize area efficiency. This type of spatial condition is

similar to the prioritizations of the MHR framework that state zones should ideally be "delineated in one piece," although this

is not a physical reality (Meybeck et al., 2013). The cumulative probability interval [0,1] was divided into 15 equal parts, each

corresponding to a separate zone, and the upper and lower bounds of ET thresholds for each zone were determined from the

CDF of mean annual ET for all global land pixels (Figure S6A). The number of ETA zones was chosen based on the number

of zones in previously established systems and the relative improvement of ET coherence with the addition of more zones

(Figure S6B).

The second proposed classification system, ET Variability-optimizing (ETV), was based on the principle of

maximizing within-zone ET coherence subject to the tradeoff of increasing complexity by adding zones. By fitting the

empirical CDF with a continuous distribution, zone boundaries can be determined analytically for a minimum desired $CV_{min}$.

For simplicity, and supported by empirical evidence (Figure S7), we fitted a uniform distribution, which is characterized by

lower and upper bounds $a$ and $b$, with $CV = (b - a)/[\sqrt{3}(b + a)]$. The ET limits defining each zone, $i$, were then determined directly from this relation as


$$a_i = \frac{b_i(1-\sqrt{3}CV_{min})}{1+\sqrt{3}CV_{min}} \qquad (2)$$

where the upper and lower limits of sequential zones are shared (i.e., $b_{i-1} = a_i$). The largest value of $b = 1{,}454$ mm yr$^{-1}$ was based on the maximum ET for all pixels, and $CV_{min} = 0.075$ was chosen based on marginal CV decrease with increasing number of zones (Figure S7-B), which resulted in 29 zones. This method produces nearly equal CV in all zones. Corresponding

ET limits for each zone are shown in Figure S7-A.

The third univariate scheme proposed here is the ET Clustering (ETC) classification system, in which the k-means clustering approach was applied. Previous analyses have used clustering techniques for climate classification purposes (Tapiador et al., 2019), including for the construction of the KHC boundaries (Knoben et al., 2018). Zones were built using a multi-start framework (n=80 starts) by forming clustering centers iteratively until the within-zone sum of squares of mean

annual ET, based on Euclidean distances, was reduced. This method encompasses aspects of both ETA and ETV, in which ET variability and area distribution are considered. The ETC approach serves to compare a clustering methodology against the previously described analytical ET-based zoning frameworks. The final number of 20 clustering centers (i.e., zones) was selected based on the smallest number of zones with CV of mean annual ET below a low threshold, selected here as 0.1 (Figure S8).

**2.6 Novel multivariate climate classification system**

The final system developed in this study is a multivariate climate clustering framework, which was created from the same k-means clustering method described for the ETC framework. This new climate classification system included two hydroclimate variables (mean annual P and PET) and was designed for comparison against the univariate ET classification frameworks, as well as previously established systems that were similarly formed from multiple variables. This final system

is herein referred to as the Water-Energy Clustering (WEC) climate classification system.

Since the KPG is the standard framework to which other systems were compared, a main objective was to create a classification scheme that was at least as good as KPG, while also using fewer biophysical parameters to draw zone boundaries. The final number of proposed WEC zones was chosen based on the common "elbow method" for visually determining the optimal number of clusters, or zones (Syakur et al., 2018). Within the context of the presently applied k-means clustering

method, the elbow method seeks to efficiently minimize the total within-zone sum of squares (TSS), such that the optimal number of zones exists where the rate of TSS change starts to decrease with the addition of more zones. According to the goal of efficiently minimizing TSS, about 5 zones would be best (Figure S9). However, the aim of this study was to optimize zones based on hydroclimate coherence and zone shape complexity. Considering this premise, the "elbow" of hydroclimate

coherence (i.e., low CV values) with respect to number of zones was between 10 and 20 zones for all hydroclimate variables,

except for Δt (Figure 2). Similarly, the elbow denoting the efficient minimization of mean number of patches across zones was

approximately 15 to 25 WEC zones, but CV of zone area was relatively constant between 10 and 30 zones (Figure S10B). A

WEC system consisting of at least 10 zones yielded mean coherence values that were better than those mean coherence values

of KPG for PET, P and Q, as denoted by dark blue dots in Figure 2. It should also be noted that although there was no number

of WEC zones that provided a lower mean number of patches than KPG, most possible numbers of WEC zones yielded mean

values that were within one standard deviation of the KPG mean (Figure S10A). Also, all possible numbers of WEC zones

allowed for a more equal distribution of zone areas compared to the CV of zone area for KPG (Figure S10B). We evaluated

both 15 and 20 possible WEC zones (compared to KPG's 30 zone system) against all climate classification systems. However,

the results for 15 WEC zones will be presented henceforth, since the K-S test showed no statistical difference in coherence nor

complexity between 15 and 20 zones (the coherence and complexity results for 20 WEC zones can be seen in Table S1).


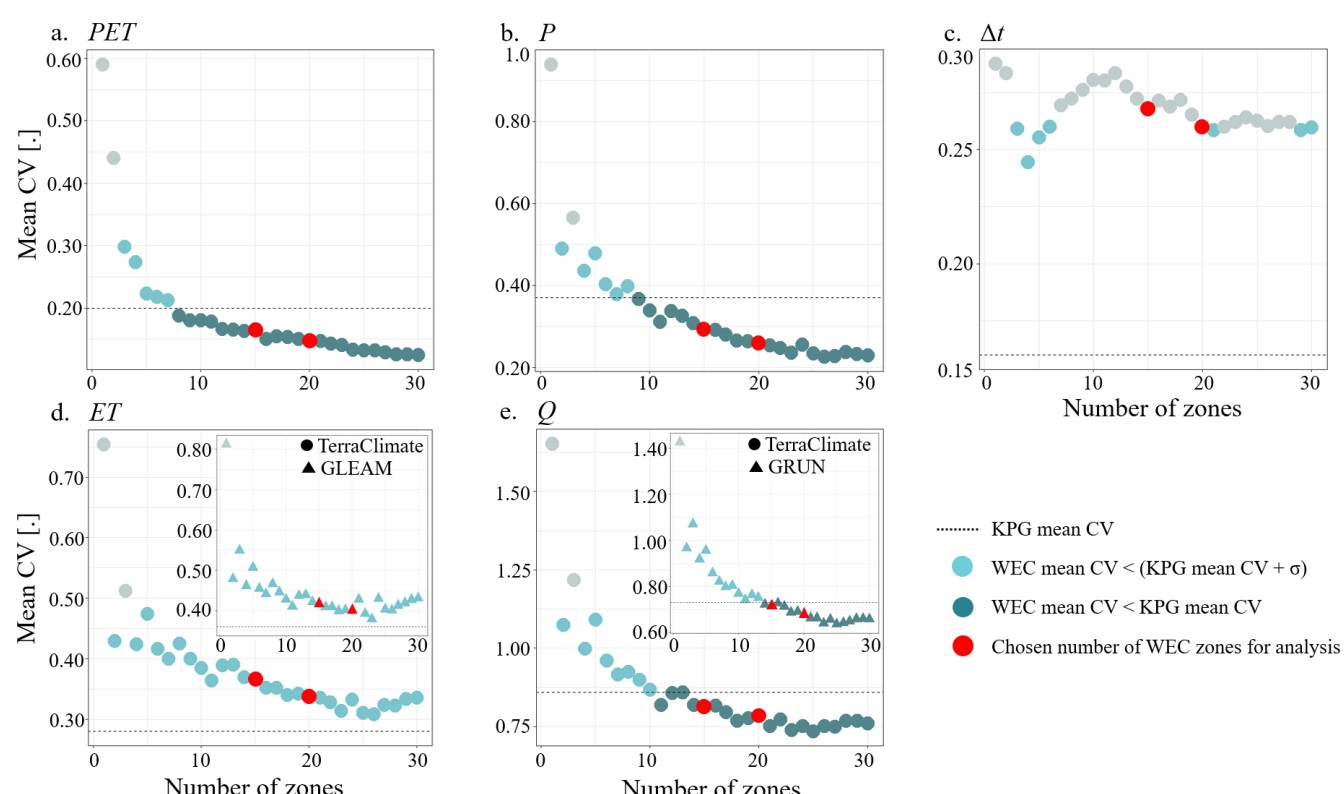

**Figure 2: Hydroclimate coherence with respect to number of possible zones within the WEC framework for all hydroclimate variables: PET (A), P (B), Δt (C), ET (D), and Q (E). For independent validation, ET and Q are also included from secondary gridded data sources, GLEAM and GRUN, respectively, and are differentially illustrated by triangles. Number of zones that yielded a mean**
**CV value lower than that of KPG (gridded horizontal line) are shown in dark blue, number of zones that yielded a mean CV value that was lower than that of KPG plus one standard deviation (σ) are shown in light blue, number of zones that yielded a mean CV value that was higher than that of KPG plus σ are shown in light grey, and the final number of zones chosen for further evaluation are in red.**

## 3 Results

This study compared four previously established climate classification systems (KPG, HDL, MHR, and KHC) and four potential new climate classification systems (ETA, ETV, ETC, and WEC) to assess for hydroclimate coherence as well as zone boundary complexities. The coherence of hydroclimate variables, PET, P, $\Delta t$, and TerraClimate ET and Q for each evaluated climate classification system is shown in Figure 3. Figure S11 illustrates the coherence of GLEAM ET and GRUN Q, which were variables used to augment independent validation. A two-sided K-S test was conducted to determine differences

between the cumulative distributions of CVs compared to a reference system for each variable. The WEC system was used as the reference system, since WEC is the novel multivariate system proposed in this study.

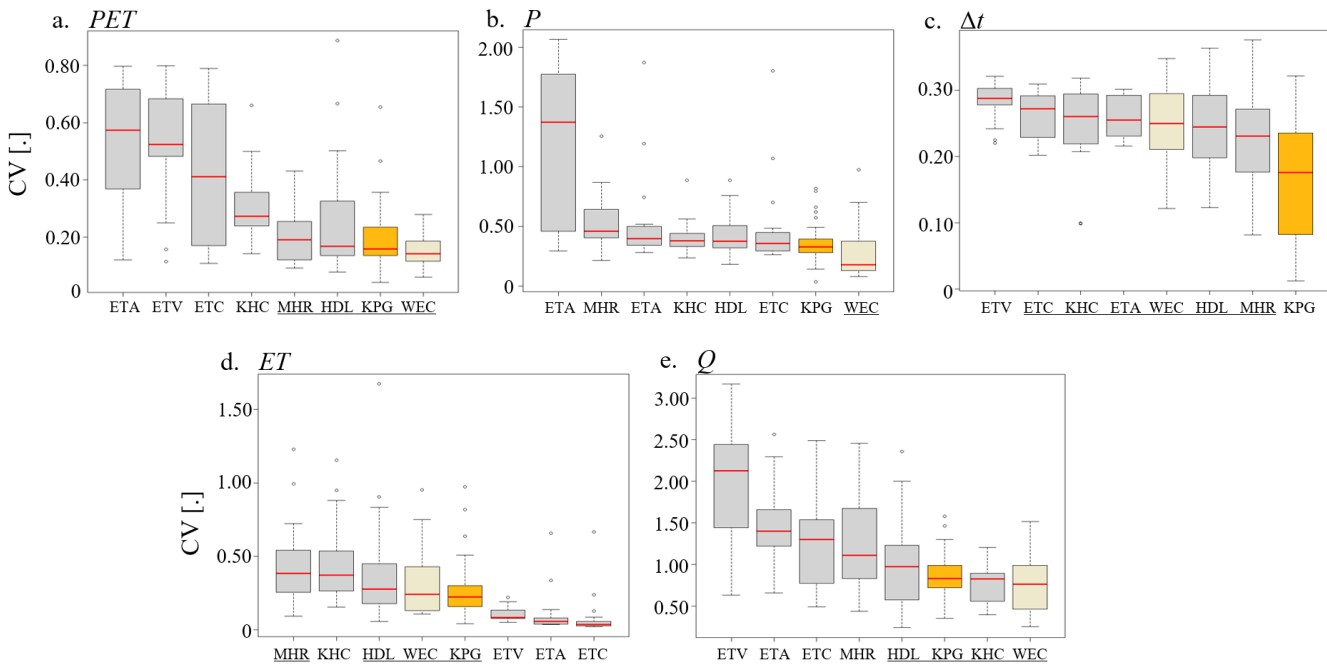

**Figure 3: Boxplots of coherence, quantified as intra-zone CV for hydroclimate variables of interest (A-E), for each assessed climate**
**classification system. In each panel KPG is shown in gold and WEC in light beige. The K-S test was used to determine whether the distributions were different from WEC. Systems whose coherence distributions were not statistically different from that of WEC are underlined.**

        Figure 4 showed MHR was the least fragmented system overall, although the KPG system also was characterized by

low patchiness when compared to the distributions of the other systems (Figure 4A). The KPG system also had relatively high hydroclimate coherence for most variables, including validation datasets GLEAM ET and GRUN Q (Figure S11), appearing as the best system (i.e., low CV values) for $\Delta t$ and not statistically different from the best system for PET and Q (Figure 3). However, it did not have the highest P or ET coherence. The high $\Delta t$ coherence of KPG is sensible, because KPG zones are built using intra-annual P and temperature (i.e., PET) dynamics. While the KPG system showed overall high coherence, which

supports its status as the most widely used climate classification system, it was not the highest for all variables, and it also exhibited high complexity with respect to zone area distribution and number of zones used in its framework (Figure 4B-C). Lastly, the number of biophysical parameters required to construct KPG zone boundaries (monthly P and temperature, n =24) is much higher than the novel systems presented here (n=2 for WEC and n=1, mean annual ET, for ETA-based systems).

The coherence of hydroclimate drivers, PET, P, and Δt, as well as hydroclimate response variables, ET and Q, were variable across systems (Figure 3). The variable that was overall least coherent was Q (Figures 3E and S11B), with both TerraClimate and GRUN Q CV values ranging beyond 1.50 for most classification systems, while the variable that was generally most coherent was Δt (Figure 3C), with CV values generally between 0.10 and 0.30 for all assessed classification systems. Of all variables, Δt yielded the greatest number of systems not statistically different from WEC, based on the K-S test for differences in CV distributions (Figure 3C). Additionally, the three novel ETA, ETV, and ETC systems had better ET coherence than the other classification systems (Figure 3D). The ETA system, along with WEC, also had the fewest number of zones and provided the most uniform zone size distribution (Figure 4B-C) but was not as coherent with respect to hydroclimate variables apart from ET (Figure 3).

The WEC system had the lowest median CV for PET, P, and both TerraClimate Q (Figure 3A-B and E) and GRUN Q (Figure S11B). It is reasonable that the WEC system is the most PET and P coherent, since these were the variables used to form the zone boundaries of the system. The high coherence of GRUN Q serves as an independent validation of the WEC framework, such that it can be concluded that the WEC system most effectively bounds zones that capture water availability drivers. Although the ET-based systems were best at bounding within-zone ET similarities and yielding high coherence, WEC did not perform worse than KPG in ET coherence, according to the K-S test (Figure 3D). The WEC system was also relatively less complex compared to most other systems, including KPG, with respect to zone area distribution and number of zones required to draw hydroclimatically coherent boundaries (Figure 4B-C). The WEC distribution of the mean number of patches in each zone was statistically different from that of KPG, and the WEC system had the next lowest median value following KPG (Figure 4A). Since the proposed WEC system had similar or better performance than the KPG system in most coherence and complexity metrics (except for Δt coherence and mean number of patches), and required 2 compared to 24 parameters to construct, the evaluated WEC framework was selected as the overall best hydroclimate classification system.

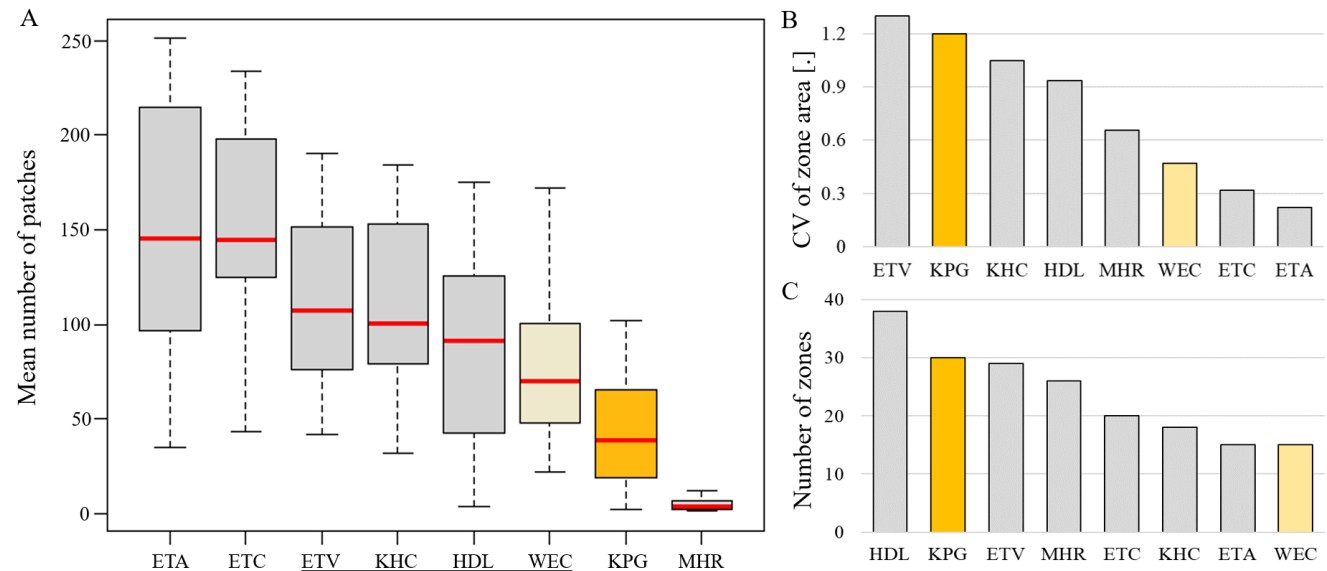


**Figure 4: Boxplots of mean number of patches per zone (A), barplots of CV of zone areas (B), and barplots of number of constructed zones (C) for each assessed climate classification system, with KPG shown in gold and WEC in light beige. The results of the K-S test were used to determine statistical difference of distributions compared to WEC. Systems whose distributions of mean number of patches were not statistically different from that of WEC are underlined.**


The KPG system qualitatively groups 30 zones into 5 primary categories (Tropical, Arid, Temperate, Boreal, and Polar), and here the 15 WEC zones were also divided into 5 primary groups by ranking zones based on increasing zone mean aridity index, $\varphi = \frac{<PET>}{<P>}$, where brackets indicate spatial average within a zone and $\bar{\varphi}$ is the mean across zones within a group. The ranked zones were evenly grouped into the five categories: Superhumid ($\bar{\varphi} = 0.39$), Humid ($\bar{\varphi} = 0.58$), Temperate ($\bar{\varphi} =$

1.07), Arid ($\bar{\varphi} = 2.05$), and Hyperarid ($\bar{\varphi} = 9.56$). Note that the single WEC zone with highest aridity encompasses the Sahara, parts of Saudi Arabia, and western Australia, for which $\varphi = 14.8$. Maps of the boundaries for the proposed WEC system and the standard KPG framework are compared in Figure 5. While there were some spatial similarities (e.g., see the Iberian Peninsula in Figure 5), most regions were divided differently. For example, parts of northern Europe were mainly divided into three KPG zones but four WEC zones. Similarly, the southeastern United States, excluding south Florida, was

mostly one KPG zone, but was separated in the WEC system into two distinct zones. The KPG framework conversely divided eastern and western Europe in respective temperate and boreal zones, while WEC treated western Europe as more heterogeneous. Clustering centers, which are the arithmetic means of each of the clusters, for the WEC climate classification system are listed in Table S2.

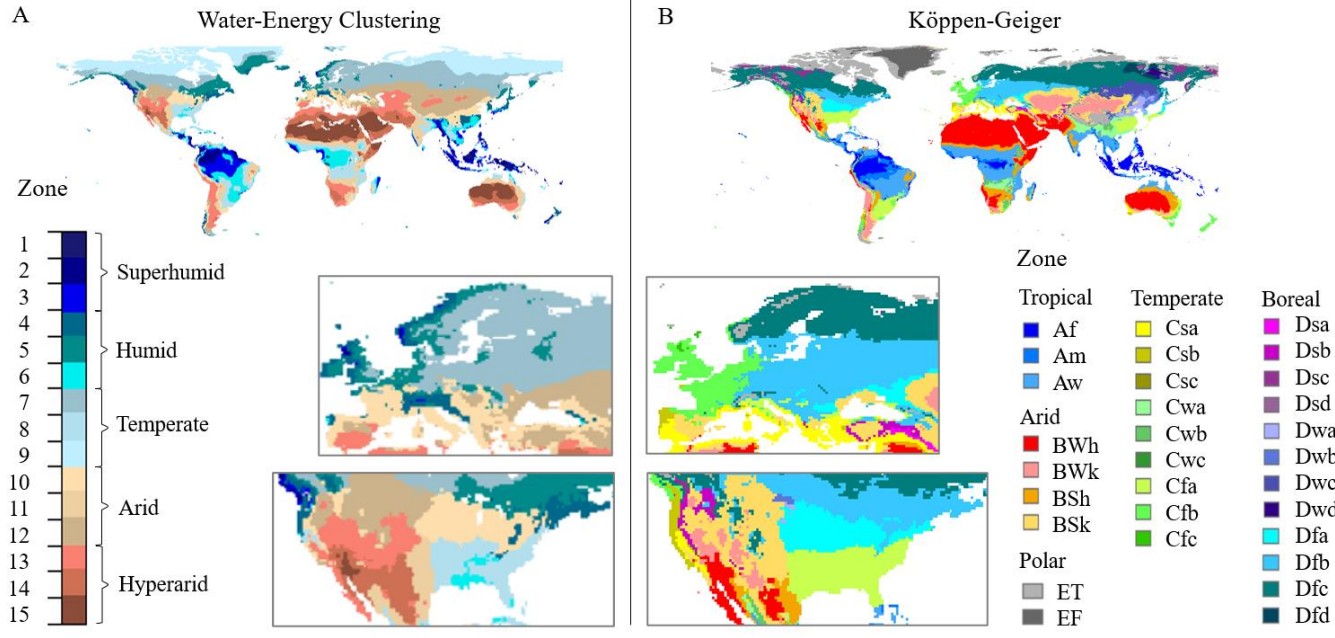

**Figure 5. Spatial distribution of WEC (A) and KPG (B) classification systems. Europe and North America are magnified.**

## 4 Discussion

We hypothesized that variable coherence and zone shape complexity would be related to the governing principles of the classification systems, which was mostly supported by the results of this study. For example, the principle of contiguity in the MHR system led to the lowest patchiness of all systems evaluated, so this system could be useful when continuous boundaries are important for ease of implementation or interpretation purposes. Additionally, concordant with the objectives of each ET-based framework, the three univariate ET-based classification systems had the highest ET coherence, while ETA (which additionally optimized equal zone area) also had the most uniform area distribution across zones. The KPG framework had the overall highest Δt coherence of the eight total compared systems, which is reasonable since KPG was the only system that accounted for monthly variability of water (P) and energy (temperature), which results in 24 biophysical parameters (Beck et al., 2018). The KHC framework similarly accounted for the long term mean monthly ratio of P and PET, but it was not particularly high in Δt coherence (Figure 3C). The WEC system was also based on water (P) and energy (PET), but from a mean annual perspective, thus requiring only two biophysical parameters as input variables. It is important to highlight that all novel systems presented here required fewer input variables, a notable aspect of system complexity, than any other evaluated previously established climate classification system, and substantially fewer than KPG.

Of the four previously established systems, KPG was the most hydroclimatically coherent but had high zone area variability (Figure S10), even with the omission of the two small KPG zones when resampled to 0.5° x 0.5° resolution in this

study. When comparing all eight systems, WEC had the highest P, PET, and Q coherence and similar ET coherence to KPG. Although it was not surprising that the WEC classification systems yielded the highest P and PET coherence, given these were the variables used to draw its zone boundaries, WEC also had much more uniform zone area distribution, half the number of zones, and required substantially fewer parameters when compared to KPG. Areas of similar water availability rates, as defined by low CV of Q, was best delineated by WEC, given that this system yielded the highest coherence for both TerraClimate Q and the independent validation source, GRUN Q. The MHR system used long term mean Q as a governing principle (Meybeck et al., 2013), while the KHC framework considered the Q regime as independent validation for their zones (Knoben et al., 2018). Although the MHR system used mean annual Q in their framework, it was not comparatively high in mean annual Q coherence, perhaps because their relatively larger zones did not reduce within-zone Q variability as much, or because the present analysis considers locally-generated Q (P – ET) and not gauged streamflow as they did. However, the KHC framework used gauged streamflow data for system evaluation (Knoben et al., 2018), and this system yielded a distribution of CV of Q not statistically different than that of WEC, which yielded the lowest median CV of Q.

Optimizing ET variability was a previously unconsidered objective in creating and validating climate classification schemes. The climate classification system comparison presented here supports the longstanding assertion that the primary mean ET drivers, water and energy (i.e., P and PET), are important considerations for broad hydroclimate analyses. To delineate the landscape based on ET dynamics, the Budyko framework is a longstanding, well-vetted mechanism for estimating the evaporative index (ET/P) using the primary drivers of the water budget, PET and P, as represented by the aridity index (Budyko, 1974; Milly, 1994; Reaver et al., 2020a; Reaver et al., 2020b; Zhang et al., 2004). We conclude that hydroclimate coherence is best achieved when P and PET are the governing principles of a zoning framework. However, when specifically evaluating ET dynamics, applying an ET-based delineation could be useful, especially if the objective of such a study is to distinctively evaluate factors that influence ET. It should be noted that boundaries created by ET drivers and not ET rates may influence the determined importance of such drivers, since intra-zone driver variability is likely to be reduced. Based on both ET coherence and spatial complexity, the ETA system established here is suggested for ET-focused questions such as large-scale assessments of ET drivers or of crop productivity (Howell et al., 2015).

This study is limited by a few factors. First, distinct climate zone boundaries, although useful in practice, do not exist in the physical system (Knoben et al., 2018). Second, this study compared averaged metrics that were applied across zones within each classification system and did not distinguish between individual zones, which could be evaluated in subsequent studies. Third, the focus on long-term mean annual hydroclimate attributes for zone formation does not account for interdecadal climate dynamics. Last, the TerraClimate ET and Q data used to assess the suite of classification systems was in part formed using the same CRU climate data used here to create the WEC boundaries (Abatzoglou et al., 2018). However, GLEAM ET and GRUN Q were also used as independent datasets and did not yield different results, which is likely due to two primary reasons: 1) the spatial scope of this analysis is sufficiently large such that calibrated rates for all hydroclimate variables are regionally representative (Abatzoglou et al., 2018), and 2) similarly, long term hydrologic dynamics are not as subject to interannual variability, since these effects are more muted across longer timescales. In this way, the broad spatiotemporal

nature of this analysis makes it reasonable that all available P, ET, Q, and PET data are appropriate metrics for forming more robust hydroclimate boundaries and subsequently assessing the water and energy budgets therein.

## 5 Conclusions

The KPG system is the most widely used climate classification system, and this analysis revealed that it indeed has relatively high hydroclimatic coherence with respect to several variables, but it also has high spatial complexity as evidenced by multiple metrics, in addition to its 24-parameter requirement. It was concluded that WEC was either better than or not statistically different from all other previously established systems, including the KPG framework, in all assessed coherence metrics apart from $\Delta t$. Moreover, compared to KPG, WEC builds half the number of zones using only two parameters as input

variables and delineates a more uniform zone area distribution to better facilitate meaningful spatial interpretations.

It is widely accepted that water and energy, chiefly in the form of precipitation and solar radiation, govern long term socioecological water availability at large spatiotemporal scales (Budyko, 1974; Berghuijs and Woods, 2016; Knoben et al., 2018; Sanford and Selnick, 2013). Several previous climate classification systems aimed to represent this water-energy interaction within bounded zones that encompass similar hydroclimatic sensitivities (Knoben et al., 2018; Meybeck et al.,

2013). It was concluded here that WEC, using water and energy in the form of P and PET rates, was the best overall system for building zones that encompass similar Q rates. This suggests that the WEC scheme is valuable for assessing and predicting water availability changes given changes in water and energy. Therefore, WEC is the most relevant system for direct management understanding and application as it relates to hydroclimate dynamics.

This study proposes WEC as a new framework for regional hydroclimate inquiries and other large spatial scale

research endeavors that may be influenced by hydroclimate systems that vary across the landscape. The WEC system is robust, since it is based on long-term mean annual rates that have low susceptibility to interannual and seasonal variability. This work is a promising pathway to regionalization within many different biophysical and socioeconomic contexts, clustering drivers to form zones of similar response variable sensitivities in order to more accurately extrapolate locally derived results and regional impacts of local management practices. The WEC framework can thus inform regional to national scale management strategies

in the effort to account for potential hydroclimate zone-dependent responses to climate and land cover changes.

**Author Contribution**

KLMP performed the analyses and led the manuscript preparation. JWJ conceived and directed the study.

**Competing Interests**

The authors declare that they have no conflict of interest.

**Acknowledgements**

This research was supported in part by USDA National Institute of Food and Agriculture Hatch project FLA-SWS-005461.

JWJ was supported in part by the National Science Foundation under award number 2000649.

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
