# Peer review of "Coherence of Global Hydroclimate Classification Systems"

_Hydrology and Earth System Sciences, 2020_

## Referee Comment (RC1) · Anonymous Referee #1 · 27 Oct 2020

The submitted manuscript suggests an innovative and parsimonious climate classification system for hydrological applications. A detailed comparison of system coherence obtained from four established and four proposed climate classification systems is provided. The proposed classification looks interesting and promising for several hydrological applications, although the paper needs some improvements before publication. In what follows, the authors may find key and minor comments.

Page 3, l. 83: How did the authors perform this? By subtracting long-term mean annual from annual values? Please add more details on this.

Page 3, l. 89: KPG - please define acronym at first occurrence.

Page 4, eq. (1): is it correct to have y over-bar, or is it y_m over-bar (see "monthly mean" as reported in l. 95)?

Page 4, l. 106: "established" rather than "veteran"?

Page 4, l. 107: I would suggest to add citations immediately after KPG and HDL.

Page 5, l. 136-137: I recommend the authors to show this uniform CDF in Figure S1, or, better, add a new figure in SI showing the comparison between empirical and analytical CDF

Page 5, l. 147: select between "means" and "k-means" and apply it consistently

Page 5, l. 150: "CV of mean annual ET" instead of "ET mean CV"

Page 5, l. 151: "system" instead of "systems"

Page 6, l. 158: Referring to zone complexity, more details on thresholds are needed. I suggest the authors to move this part from SI to the main paper and add a discussion.

Page 6, l. 159: in SI, coherence is multiplied by 1.50. Are the authors assuming that coherence in WCE system can be larger or equal than KPG plus 50% KPG? If this is the case, please clarify this in the main text and also in SI. In SI, numbers in squares are not clear. If it is a product, simply add a dot between numbers.

Page 6, l. 162: From SI: "Hierarchically, water budget coherence and number of zones were given highest priority. Therefore, the P,PET clustering system with 22 zones (denoted Water-Energy Clustering), was chosen for comparison against the other climate classification methods." How do the authors choose this? The authors should better explain this fundamental part in the main paper and also add more details on the sensitivity analysis performed for the number of zones.

Page 8, ll. 197-203: only the last sentence seems to be reasonable. Other comments try to justify the definition of ETA and ETC systems and support their performances, but both ETA and ETC show similar performance as MHR and KHC. Actually, except

for CV(ET), even the proposed systems show similar performance compared to established systems MHR and KHC. This was somehow expected since the authors defined ET-based systems. I would suggest to improve this discussion by highlighting that WEC is the best model from the new ones. "similar P coherence to KPG": I cannot see this from Table 1, where CV(P) for KPG=0.38, ETA=0.56, ETC=0.47.

page 8, l. 215: please explain what phi over-bar means

Page 8, ll. 216-217: the authors are invited to show this in SI. Actually R2=0.25 is very low.

Page 10, l. 235: "Discussion and conclusions" instead of "Discussion"

---

## Referee Comment (RC2) · Anonymous Referee #2 · 29 Oct 2020

Review of "Coherence of Global Hydroclimate Classification Systems" by Pisarello and Jawitz

**Summary**

The authors collect four existing climate classifications and create four new ones. The four new classification schemes are all some variety of dividing regions into groups with low internal variability of actual evaporation (ET) or precipitation and potential evapotranspiration (P & PET) rates. The eight classification schemes are then evaluated on their ability to produce coherent (little variability within each climatic group) and spatially non-complex hydroclimatic groups. Hydroclimatic coherence is assessed on ET, P and PET data (which were also used to create the four new classification methods), $\Delta\theta$ values (a measure of temporal alignment of P and PET seasonality), and modeled Q data. Spatial complexity is defined as the number of groups in the classification, if these groups are of equal size and if they are connected in space. The authors find that one of their own classification schemes (called WEC) performs best on these criteria and recommend it to others.

I have read this paper with interest and I think (hydrologically-informed) climate classifications can and should be further developed. I have however various serious concerns about the experimental design the authors present in this manuscript and about the presentation of the work in general. Briefly, I think the authors can do a much better job in explaining how their preferred climate classification scheme was developed. More importantly, I believe that their experimental design makes it a foregone conclusion that the proposed WEC scheme beats the other classification approaches, because the WEC setup uses the same data as are used for WEC evaluation. The manuscript also lacks in clarity. Especially the setup of WEC is difficult to understand. I have outlined these concerns in more detail below.

**Major**

I here summarize my main concerns about the experimental design. Line-by-line comments are provided below. There is a certain amount of duplication between this section and the line-by-line comments.

1. It is unclear how ETA, ETV and ETC are different in concept
   The authors propose three mono-variable clustering approaches, based on ET. To me, these seem like the same approach with only very minor differences in underlying details. I recommended that the authors make clearer why these minor differences are enough to treat these three schemes as completely independent. More details in line-by-line comments below.

2. The WEC description is too brief
   I struggled to understand exactly how WEC is set up and am left with many questions after reading the Supporting Information to this paper. I strongly encourage the authors to provide more specifics about their methods, both to let the reader better understand the results presented in this paper and to allow the reader to reproduce the WEC results if they wish. More details in line-by-line comments below.

3. Evaluation data is not independent from the data used to set up the four new schemes
   The authors use data from the CRU TS data set (P and PET) and simulations from the TerraClimate data set (ET) to setup the four new classification schemes. This same data (P, PET

and ET) is also used for 3 out of the 5 climate coherence evaluation criteria, where all eight schemes are compared on their ability to group this data into groups with little within-group variability. Unsurprisingly, the four new schemes that are all explicitly conditioned to create such groups with that particular data do very well in this assessment. This comparison is meaningless and should be removed from the manuscript. More details in line-by-line comments below.

TerraClimate simulations of streamflow (Q) are also used to evaluate the eight classification schemes. These Q simulations are the result of forcing a very simply hydrologic model with P and PET data which, critically, are in part obtained from the same CRU TS dataset that is used to create the four new classification schemes. The TerraClimate ET and Q data can not be seen as independent of the data used to setup the WEC scheme and this undermines the conclusion that WEC is the superior climate classification scheme. If the authors are seeking to thoroughly evaluate the capability of their cluster-based classification, independent evaluation data is needed. This might be obtained from observations of streamflow (instead of model results, the global GSIM database might be of use) or actual evaporation rates generated by modern land-surface schemes. The simple model that underlies the TerraClimate ET and Q values has been first used in 1948 and hardly modified since, and I expect that by now something better may be available. More details in line-by-line comments below.

4. The spatial complexity criteria are not well justified and possibly need changing or removal
The authors define criteria that reward classification schemes that return large, connected zones with a single internal climate and uniform sizes. However, the main aim of climate classification schemes is to find locations that have similar climates, not necessarily locations that are spatially connected to one another. Neither do I see much reason to assume that all climate groups must be of equal size. These must be better justified or removed. More details in line-by-line comments below.

5. KHC conversion to discrete categories is too simplistic
The authors convert the continuous climate classification by Knoben et al. (2018) into a discrete climate classification for comparison purposes. Rather than using a clustering approach as is done in Knoben et al. (2018), the authors instead "round pixel values until 30 climate classes are created". This sounds somewhat simplistic to me and I recommend instead that the authors use some form of clustering to overlay separate climate classes over the continuous hydroclimatic spectrum (KHC). I suspect that this rounding procedure is one of the main causes why KHC shows very high numbers of patches in this study and why Koppen-Geiger (KPG) and KHC appear equally coherent on the Q variable in this study, whereas KHC is substantially better than KPG for finding similar Q regimes according to the assessment in Knoben et al. (2018). More details in line-by-line comments below.

6. The discussion is somewhat limited
I think the discussion would be stronger if the authors address the question "what did we learn about the world?" from their analysis. More details in line-by-line comments below.

**Line-by-line comments**

L52.    It is equally possible that actual ET rates are hardly used in classification schemes because observations of actual ET are hard to come by at the spatial scales where climate classification is typically useful. Global products of actual ET are typically the result of model simulations or derived from model simulations across this domain. Depending on the origin of the actual ET data, one could argue that a classification that uses this kind of data is more of a modeled-climate classification and is critically reliant on the accuracy of the model simulations with respect to actual (meaning occurring in reality) ET.

L82.    The provided reference seems to say that TerraClimate data are at a monthly resolution. If this is the case, how was this disaggregated into daily data?

L89.    KPG has not been defined yet.

L95.    Should "monthly mean" be "annual mean"?

L101.   Mentioning just the mean $R^2$ values and their standard deviations seems quite a short description of the accuracy of these fits over 61000+ grid cells on the planet. A somewhat longer description is appropriate. Do the authors' results for P match those of Berghuijs & Woods (2016) in terms of accuracy and spatial patterns of high/low accurate fits?

        In which regions are the authors' fits more/less accurate?

L107.   "veteran" seems an odd choice of words. Maybe "legacy"?

L111.   It is not entirely clear to me how pixel values can be rounded into creating a distinct number of categories. Clustering of the KHC climate index values (similar to how WEC is created) would be much more appropriate and is also the approach taken in Knoben et al. (2018) to create discrete categories that overlay their hydroclimatic continuum. See Figure 3 in Knoben et al. (2018). Can the authors clarify their rounding approach and justify why they use this over a clustering approach that uses the actual data of the KHC scheme?

L122.   I don't quite follow the arguments presented here that should support the idea of a classification that has an equal number of pixels in each zone and I think these arguments need to be clarified or changed. See my concerns with each individual argument below.

        #1. Koppen-Geiger (K-G) has "relatively high spatial non-uniformity […] resulting in highly variable relevance for regional analysis." I interpret this as meaning that K-G has climate classes of non-uniform size and that this is not useful if one's region of interest falls entirely within a single K-G class out of the ~30 or so possible K-G classes. I don't quite understand how dividing the entire globe into only 15 classes addresses this problem, as using 15 classes compared to 30 necessarily means lower granularity in the authors' ETA scheme than is possible with K-G. Additionally, I see no reason to assume that forcing each of the 15 ETA classes to have an equal number of pixels will necessarily mean that "regional" analyses have more useful climatic information available. If we consider 5 continents globally, this scheme roughly divides each continent into 3 classes. Does this really help regional studies?

**2. "Additionally, it is useful to have a simple baseline framework upon which to compare other systems." I agree in principle, but I'd argue that even a simple baseline needs to be somewhat plausible. Dividing a map into 15 regions of equal size, based on a single variable seems a pretty low threshold to beat. Why not use an existing classification as a benchmark?**

**3. "Zones should ideally be delineated in one piece." This seems somewhat counter-intuitive to me, because the express purpose of climate classification schemes is to identify regions that are similar in terms of variables X, Y and Z. Whether such regions are spatially connected is irrelevant. This also directly c argument #1, where having large areas of a single climate class is mentioned as a negative aspect of the K-G scheme.**

L124.   It is unclear to me how the proposed ETA scheme should be interpreted in a temporal sense. Given that the number of pixels in each class should be the same, this means that the resulting 15 classes are only valid for a given snapshot in time. If the chosen time period for this classification changes, the underlying ET data would change, and wouldn't therefore the number of pixels in each class change too? If we do not want to violate the "equal number of pixels in each class" concept, this means that classes need to be redefined when the underlying time period changes and thus the classes do not have a consistent meaning for different time periods. Imagine a theoretical case where PET uniformly increases over the planet with a constant value. The lowest ETA class now corresponds to a very different real-world climate than it did before.

L124.   Using number of pixels is not necessarily a way to guarantee a relatively even distribution of zones in terms of area (which the name "ET Area-optimizing hints at). Based on the CRU data, each pixel represents a certain area on a regular latitude/longitude grid. Pixels in such a grid do not translate easily into real-world areas. A single 0.5x0.5 degree pixel at the equator might be approximately 50 km^2, while the same pixel size near the poles would represent a fraction of that area.

Can the authors clarify why having even distributions in the number of pixels is desirable, even if this could potentially lead to a very uneven distributions of zone size in km^2?

L133.   I don't quite understand what makes ETV different from ETA. ETA imposes groups on the empirical ET CDF. As a result, each ETA group consists of regions with similar ET values (i.e. low CV within the group).

ETV seems to minimize within-group CV of ET values by imposing groups on a normal distribution fitted to the empirical CDF. It seems to me that the only difference between ETA and ETV is that ETA uses the simulated ET values (from TerraClimate) directly, whereas ETV uses an approximation of these ET values.

ETA with 10 groups has CV = 0.2 (Fig S1B); ETV with 10 groups has CV = 0.2 (Fig S2). What does the fitted normal distribution add to this analysis that makes ETV with 29 groups (as determined on line 144) substantially different from ETA with 29 groups?

L136.   Is "S1" the correct cross reference? I don't see a fitted cont. uniform distribution in Fig S1.

L144.    Why did the authors choose to use 29 zones? To me it currently sounds that to maximize within-zone ET coherence, one would simply keep imposing more groups until each zone contains a single pixel and within-zone ET CV equals 0.

L145.    I'm again a bit confused about the difference between this approach and the preceding ones. Constructing an (empirical) CDF already puts locations with similar mean ET values close to one another, which is also what the clustering in ETC tries to achieve.

Additionally, because global mean ET values are approximately continuous (as evident from Fig S1B, S2 and S3), K-means will be trying to impose distinct boundaries on continuous data and therefore tend to gravitate towards clusters of approximately equal size. The authors have already defined ET zones of equal size with minimal within-zone ET variability in their ETA approach. So what does using a clustering algorithm add?

Equally, comparing Figure S2 and S3 seems to show that ETV and ETC generate approximately the same CV for the same numbers of clusters/groups, but with some scatter in the ETC values (potentially caused by the initial guesses for cluster centroids, see comment below).

L146.    K-means clustering is quite dependent on the initial guess of cluster centroids for the location of the final clusters. A multi-start framework shows to what extent this vulnerability influences the final clusters. Was the K-means algorithm used in a multi-start framework? If not, why not?

L150.    Is there any particular reason why CV = 0.1 makes a good threshold?

L151.    Should "systems" be "system"?

L162.    I find the description of this new classification scheme in the SI too brief to understand in detail what's going on. I gather that the authors used K-means clustering on various combinations of data but the rest of the method escapes me. This must be addressed, because it (1) makes the authors' claims that WEC is the best out of the 8 classification schemes difficult to assess; and (2) makes the classification impenetrable to others who might wish to reproduce or use this work.

Some of the questions I currently have upon reading the SI:

1. How were the combinations of P, PET, ET, Q and delta theta determined? I notice that not all possible combinations are present in Table S1.

2. What were the K-means settings? Was the algorithm restarted multiple times to test cluster stability?

3. How were the thresholds for coherence chosen (SI, page 3)?

4. What does (1.50) refer to in "KPG(1.50)"? Is this the 50% deviation mentioned in the main text? If so, this should read "KPV value * 1.5". Also, the use of "=" signs is extremely confusing in the lists on this page and should be removed.

5. Why does the reader need to know the KPG coherence scores at this point in the analysis?

6. Is "number of parameters" equivalent to number of K-means clusters? If not, which K-means parameters are meant?

7. If number of parameters is not the same as number of clusters, how was the appropriate number of clusters chosen?

8. What does "number of patches" refer to? How was its threshold determined?

9. I don't understand how/why the P,PET clustering system was chosen out of all possible options, nor why 22 zones are considered appropriate (SI, page 3).

10. The caption of Table S1 states that cells with a "1" in them indicate a combination of variables and number of clusters that meet the criteria specified on page 3 of the SI. From Table S2 however, it seems that neither ET,PET not P,PET meet the "number of patches" criterion. Why are these then shown in table 1 as if they do meet all criteria?

11. Similarly, neither ET,PET nor P,PET meet the CV(ET) criterion (all values are > 0.33), and all but one fail the CV(Q) criterion (only P,PET with 22 zones has CV(Q) < 1.31). Why does table S1 show these results as meeting the criteria? Why define criteria at all if they are not used?

12. I don't understand why there are gaps in Table S1 between certain rows with "1" in them. For example, column P,PET. If 22 clusters are sufficient to meet all criteria (indicated by a "1"), and all criteria are aimed at minimizing differences within a single cluster, it is logically impossible that using 23-27 clusters gives worse results than using 22 clusters, especially considering that 28 and 29 clusters suddenly do meet all criteria again. The only explanation I can think of is that certain settings in the K-means algorithm prevent it from finding the most optimal cluster configuration when 23-27 clusters are used. The multi-start issue I have mentioned before is a possible culprit.

13. Why were ET,PET and P,ET chosen for further analysis in Table S2 and not others?

L158.    "Zone complexity" is undefined up until this point and the reader can only guess at what this means by reading the SI. I suggest to clarify what is meant by this in the main text and to also explain whether it is low or high zone complexity that is desirable.

L160.    The SI could use a header to indicate where this section starts.

L164.    This section seems to be the justification for many of the authors' methodological choices, in particular about their selection of classification schemes. I suggest to move section 2.6 to the beginning of section 2, so that reader already has access to this information before it is needed to understand the authors' methodology.

L168.    The authors argue that "classification systems should consist of a relatively even distribution of pixels across zones, avoiding disproportionally large or small zones" (similar to an earlier argument in section 2.4). I don't understand this argument for two separate reasons:

1.    I don't think there is much reason to assume that climatic zones should follow an even distribution across the planet, either in terms of pixels or in terms of area. Globally, deserts are big and alpine regions are relatively small. A classification scheme that tries to create climatic zones of equal (pixel) size will not capture either climate properly, and thus offer little in terms of hydroclimatically relevant information.

2. Like I argued before, I don't know if number of pixels is an appropriate unit here. KPG polar zone ET might take up a fair number of pixels on a regular grid, but in terms of total area the arid classes dominate (compare Sahara size is ~9.2 million km^2, whereas Greenland is ~ 2.2 km^2). I don't think pixels are a particularly helpful unit for this analysis.

I recommend to remove this criterion from the analysis.

L169. The authors argue that "Zones should be as hydrologically continuous as possible (Meybeck et al., 2013), minimizing patchiness or fragmentation". Like I argued earlier, this is counter intuitive to me. Climate classification systems are intended to find places that are similar climatically, regardless of physical distance. By penalizing systems for patchiness, the authors effectively favor classification schemes that generate large areas of single climate zones, without providing any justification that such schemes are more representative of the real world. In effect, the less data a scheme uses, the more likely it is to generate large connected areas of climate zones and thus, according to this criterion, the better this scheme is. This seems extremely counterintuitive to me and I recommend to remove this criterion and to re-do the analysis.

L185. Like before, I'm really not sure why different numbers of pixels contained in different climate classes is a bad thing. If different climate types cover differently sized areas on a map, than that's simply how the global hydroclimate is. Re-drawing the climate class boundaries to create equally large zones is not adding any new hydro-climatic insight to the problem (in fact, one might argue that such an approach uses *less* hydro-climatic insight).

L204. Seeing the authors' assessment scheme indicates some serious methodological concerns, centering around the fact that no real independent evaluation data has been used.

1. Classification schemes are evaluated on their ability to create low within-zone variability of ET values (i.e. low CV(ET)). This is the exact same data that has been used to condition the authors' ETA and ETC schemes on and, unsurprisingly, when one specifically sets out to create groups with as low as possible within-group ET variation and then uses that same data to see how well that worked, these schemes are impossible to beat. In my opinion the CV(ET) comparison is meaningless because it cannot reasonably be expected that any scheme beats the ETA and ETC schemes in this comparison.

2. The same argument can be applied to the CV(P) and CV(PET) criteria. The authors' WEC scheme is specifically conditioned on creating groups with low variability in these two climate indicators and therefore the comparison with existing classification schemes is meaningless. The only question this seems to answer is "are established climate classification schemes better at clustering P and PET values then a clustering algorithm can cluster P and PET values?" Apparently and not entirely unexpectedly, they are not.

3. The ET and Q data in this study are taken from the TerraClimate dataset (Abatzoglou et al., 2018). To quote directly from Abatzoglou et al. (2018):

*"A one-dimensional modified Thornthwaite-Mather climatic water-balance model (WBM)[22,31] was used to calculate monthly water balance from 1958–2015. The WBM is a single bucket model applied consistently across global land surfaces that operates on a*

*monthly time step and considers the interplay between precipitation, $ET_0$, as well as soil and snowpack water storage. The WBM accounting scheme considers runoff as the excess of liquid water supply (precipitation and snowmelt) used by monthly $ET_0$ and soil moisture recharge. Soil water is extracted during months where $ET_0$ exceeds liquid water supply, with the extraction efficiency of soil water declining exponentially with the ratio of soil water to extractable soil water capacity. Under such conditions, actual evapotranspiration is counted as the liquid water supply plus the soil water utilized and climatic water deficit is the difference between $ET_0$ and actual evapotranspiration."*

Due to its simple design and monthly time step, this model has little to no capacity to generate non-linear (and thus realistic) hydrologic behavior. It is thus likely that these simulated Q and ET data are strongly correlated to the forcing data used to generate them. TerraClimate uses the CRU TS v4.0 as one of its inputs, while CRU TS v4.0.4 is used by the authors to provide the P and PET data for their classification. WEC is thus conditioned on a dataset that is very similar to the dataset used to generate the ET and Q data that are used to evaluate the different classification schemes in this paper. It is therefore not entirely surprising to see WEC perform (reasonably) well on the CV(ET) and CV(Q) criteria. In a comparison such as this, independence of the evaluation data is critical to guarantee a fair comparison. I suggest to replace the evaluation data for Q with observations from for example the global streamflow attributes dataset GSIM, or those from the Global Runoff Data Centre. ET might be obtained from modern land surface models run at a global scale, instead of from a bucket model first conceived of in 1948 and hardly changed since.

4. As argued before, neither having equal numbers of pixels in each zone nor having a low number of patches seem particularly indicative of a useful climate classification scheme to me. I suggest to remove these if they cannot be better justified.

5. The fact that ETA has "the fewest number of zones and very high coherence for zone size (by far the lowest CVz)" (L200) should not come as a surprise when the entire premise that underlies ETA is dividing the available number of pixels in a small number of zones of equal sizes.

6. It is odd to see that ETA has the best score for "zones" (indicated in bold), because it has the lowest number of zones. This is counterintuitive to me, because this seems to favor climate classification schemes that are explicitly not good at their intended purpose, namely to define different climatic regions. The ideal score on this assessment would be obtained by a scheme that returns 1 climate zone, which means that all climates are classified as being the same. I recommend to remove this criterion.

7. Concluding, I believe that the evaluation criteria heavily favor the authors' classification schemes and that the fact that WEC performs well on them is no more than an artifact of the experimental design. The evaluation data cannot be considered independent from the data used to generate WEC (and ETA, ETV, and ETC) and the pixel, patches and zones criteria are not well-justified. This undermines the authors' conclusion that WEC is a better

classification scheme than Koppen-Geiger or any of the other existing classification approaches. Independent evaluation data is needed.

L205.   The caption mentions that certain schemes have statistically similar results. How was this determined?

L205.   I find the number of patches for KHC extremely high. Can this be a consequence of the "pixel-rounding" procedure described on line 111? This result indicates that this rounding procedure is unnecessarily crude, because the KHC groups shown in Knoben et al. (2018) are not nearly as fractured as this. Given the authors' use of actual clustering algorithms in this paper, applying those to the KHC data seems feasible.

L214.   This grouping procedure needs to be explained in more detail than is currently done. Is this the result of another k-means clustering exercise? Is phi_bar the mean aridity over the group?

L216.   Where are these relationships shown? How do these compare to those from Koppen-Geiger classes?

L217.   "… indicating higher spatial variability (lower coherence) with increasing aridity." I'm not entirely sure how to interpret this statement. Which conclusion should the reader draw here?

L226.   This discussion seems to imply that WEC provides higher granularity in certain areas. I think this argument can be reversed too: whereas KPG shows some longitudinal diversity in climate zones, WEC lumps London, Amsterdam, Berlin, Warsaw, Moscow and a substantial part of the Russian Federation into a single climate zone. I think a more balanced discussion of WEC is needed.

L235.   "Discussion and conclusions"

L238.   This statement is a bit inaccurate, because Knoben et al. (2018) use the monthly ratio between P and PET as one of their predictors.

L243.   It needs to be clarified what is meant by "parameters" here. This seems to say that the parameters used by WEC are mean P and mean ET. What about the number of clusters used and other k-means settings?

L243.   "a notable aspect of system complexity" This argument is unclear. Why is the number of parameters relevant in a climate classification scheme? With the definition of "parameters" as "number of climatic variables used" (as I gather from this section), this sentence seems to imply that the less data used, the better the classification scheme is. Can this be clarified?

L245.   It should be no surprise that WEC, as the result of a k-means clustering algorithm applied to P and PET data, consists of groups that have low internal variability on that same P and PET data. Also, Table 1 seems to say that KPG scores better on CV(ET) and CV($\Delta\theta$) while WEC scores higher on CV(Q) only, albeit with a higher standard deviation than KPG gets.

L250.   I find this odd. As the authors say (lines 48-49), Knoben et al. (2018) compare their classification scheme to KPG and find that KHC is better for hydrologic purposes (in this case for grouping locations with similar hydrologic regimes expressed through a variety of metrics). In the authors' words, KHC is "more coherent with respect to Q" then KPG. Why does this not show in the

authors' assessment? This could be related to "pixel-rounding" or to the use of simulated Q in this study.

L250. Additionally, if KHC is not high in Q coherence, why is it indicated as one of the three best systems in Table 2?

L252. I suggest to remove these statements because they are trivial. If one clusters ET data one gets groups with high ET coherence. If one divides all data into 15 equal-sized groups, one gets groups with equal sizes. This should not need to be presented as a benefit of these approaches.

L257. Which variables do the authors mean when they refer to "water budget dynamics"?

L265. I don't quite follow this sentence, perhaps because I don't understand which variables are meant with "water budget components".

L283. Given all the methodological concerns that need to be addressed, I do not think this conclusion is currently well supported.

L289. This seems overly optimistic to me. Which kind of environmental management decisions can be made at a scale where a handful of Koppen-Geiger classes or WEC groups provide a useful indication of local conditions?

L289. What I miss in this discussion is a critical assessment of hydroclimatic understanding and how the authors' proposed WEC adds to this. Existing classification schemes are based on hypotheses about how the world works and about which elements of the global climate are first-order controls on the resulting hydroclimate. WEC is simply a clustering method that finds regions with similar P and ET values. What does this teach us about global hydroclimatic relationships? If WEC is better than established methods that do rely on theory, then where is this theory faulty or incomplete? In other words, I think this discussion would be much stronger if the authors where to consider the question "what did we learn about the world?"

---

## Referee Comment (RC3) · Anonymous Referee #3 · 8 Nov 2020

The authors highlight the absence of a proper ET-derived classification system. It is a such that they propose a "series of ET-based global classifications that should yield comparatively higher ET coherence than other systems", by assessing coherence and shape complexity within the classifications. I find that the study is well carried out, very nicely written and illustrated. Methods are also well explained. I think that the study would be a great contribution to HESS, enabling researchers to choose the best system of classification suited to their purposes, specially regarding ET. However, I have some clarifications that need to be added to the former version of the manuscript.

1. Existing so many proxies for landscape connectivity assessments the authors need

to support the use of "zone area" and "zone fragmentation". Why are these the best ones? Also, I did not see any formula, or explanation. This is needed wince these classification schemes are not that well known in the field of water resources, and rather in the field of ecology. A figure would be useful. 2. ET is dependent on many local parameters related to land cover and land use (See for example Sterling et al., 2013). Human activities such as agriculture, urbanization, deforestation, heavily affect these parameters and then would imprint less coherence, more variability and patchiness into the classification system. The authors should comment/adjust on this. 3. Furthermore, I would have done the analysis with a more "large-scale climatic parameter" that involves less spatial variability (and more spatial coherence-less CV) at the local scale, such as the aridity index (PET/P) or evaporative ratio (ET/P). Have the authors considered this? 4. I was expecting a more concrete recommendation on the best system for ET. Which one is it if you have to choose one? 5. Conclusions are missing, and should be independent from the Discussion.

References: Sterling, S.M., Ducharne, A., Polcher, J., 2013. The impact of global land-cover change on the terrestrial water cycle. Nature Climate Change 3, 385–390. https://doi.org/10.1038/nclimate1690

---

## Author Comment (AC1) · 31 Dec 2020

We thank the reviewer for these helpful comments. The Reviewer comments are listed below, along with our response to each. Most comments require only minor edits. In some cases, we describe revisions to the manuscript (with line numbers), and we recognize that the revised manuscript is requested in a subsequent step.

REVIEWER 1

The submitted manuscript suggests an innovative and parsimonious climate classification system for hydrological applications. A detailed comparison of system coherence

obtained from four established and four proposed climate classification systems is provided. The proposed classification looks interesting and promising for several hydrological applications, although the paper needs some improvements before publication. In what follows, the authors may find key and minor comments.

Page 3, l. 83: How did the authors perform this? By subtracting long-term mean annual from annual values? Please add more details on this.

Response: We thank the reviewer for calling attention to this. As suggested by the reviewer, we have added clarification (line 103).

Page 3, l. 89: KPG - please define acronym at first occurrence.

Response: Corrected (line 33).

Page 4, eq. (1): is it correct to have y over-bar, or is it y_m over-bar (see "monthly mean" as reported in l. 95)?

Response: We agree with the reviewer that this was not precise, and we have clarified the meaning of y over-bar in line 116.

Page 4, l. 106: "established" rather than "veteran"?

Response: We thank the reviewer for this suggestion. "Legacy" is now used in line 136 (note that "established" was used already in the previous sentence).

Page 4, l. 107: I would suggest to add citations immediately after KPG and HDL.

Response: Corrected (line 136).

Page 5, l. 136-137: I recommend the authors to show this uniform CDF in Figure S1, or, better, add a new figure in SI showing the comparison between empirical and analytical CDF

Response: As suggested by the reviewer, we updated Figure S2 in SI to show the comparison between the empirical and analytical uniform CDF.
Page 5, l. 147: select between "means" and "k-means" and apply it consistently

Response: We have elected to retain "k-means" as this is the name of the clustering approach, while "means" refers to input variables and/or variables assessed for coherence. The respective usage of each term is consistent with those definitions.

Page 5, l. 150: "CV of mean annual ET" instead of "ET mean CV"

Response: Corrected accordingly (line 189).

Page 5, l. 151: "system" instead of "systems"

Response: Corrected accordingly (line 190).

Page 6, l. 158: Referring to zone complexity, more details on thresholds are needed. I suggest the authors to move this part from SI to the main paper and add a discussion.

Response: As suggested by the reviewer, we have updated this in the main text (section 2.6, lines 196-202) and SI (under added section "Multivariate climate classification system selection") to enhance clarity.

Page 6, l. 159: in SI, coherence is multiplied by 1.50. Are the authors assuming that coherence in WCE system can be larger or equal than KPG plus 50% KPG? If this is the case, please clarify this in the main text and also in SI. In SI, numbers in squares are not clear. If it is a product, simply add a dot between numbers.

Response: As the reviewer suggests, this has been corrected.

Page 6, l. 162: From SI: "Hierarchically, water budget coherence and number of zones were given highest priority. Therefore, the P,PET clustering system with 22 zones (denoted Water-Energy Clustering), was chosen for comparison against the other climate classification methods." How do the authors choose this? The authors should better explain this fundamental part in the main paper and also add more details on the sensitivity analysis performed for the number of zones.

Response: As suggested by the reviewer, and similar to the previous comment, this has been updated in the main text (section 2.6, lines 205-210) and SI (under added section "Multivariate climate classification system selection") to enhance clarity.

Page 8, ll. 197-203: only the last sentence seems to be reasonable. Other comments try to justify the definition of ETA and ETC systems and support their performances, but both ETA and ETC show similar performance as MHR and KHC. Actually, except for CV(ET), even the proposed systems show similar performance compared to established systems MHR and KHC. This was somehow expected since the authors defined ET-based systems. I would suggest to improve this discussion by highlighting that WEC is the best model from the new ones. "similar P coherence to KPG": I cannot see this from Table 1, where CV(P) for KPG=0.38, ETA=0.56, ETC=0.47.

Response: As suggested by the reviewer, the language in the text has been changed to more definitively reflect that WEC is the best system (lines 235-236). However, the statements preceding this conclusion that were questioned by the reviewer are supported by K-S tests, as noted in the Table 1 heading and as now added to line 232. The values cited by the reviewer are the means for CV(P), but as shown in table 1, the standard deviations of the distributions are relatively large for CV(P), resulting in statistically similar values for all methods except WEC, again as indicated by bold in Table 1 (corresponding to K-S test results).

page 8, l. 215: please explain what phi over-bar means

Response: This has been corrected in line 247.

Page 8, ll. 216-217: the authors are invited to show this in SI. Actually R2=0.25 is very low.

Response: This has been removed.

Page 10, l. 235: "Discussion and conclusions" instead of "Discussion"

Response: We thank the reviewer for calling attention to this. This is corrected.

---

## Author Comment (AC2) · 31 Dec 2020

We thank the reviewer for this very detailed review and for these helpful comments. The Reviewer comments are listed below, along with our response to each. Our main revision is that we have followed the Reviewer suggestion to update our method for converting KHC to discrete categories. This resulted in specific values changing in our results table – but, critically, the overall results were not substantially altered. We also added several clarifications suggested by the reviewer. In some cases below, we describe revisions to the manuscript (with line numbers), and we recognize that the revised manuscript is requested in a subsequent step.

[Figure]

Reviewer 2

Summary The authors collect four existing climate classifications and create four new ones. The four new classification schemes are all some variety of dividing regions into groups with low internal variability of actual evaporation (ET) or precipitation and potential evapotranspiration (P & PET) rates. The eight classification schemes are then evaluated on their ability to produce coherent (little variability within each climatic group) and spatially non-complex hydroclimatic groups. Hydroclimatic coherence is assessed on ET, P and PET data (which were also used to create the four new classification methods), $\Delta\theta$ values (a measure of temporal alignment of P and PET seasonality), and modeled Q data. Spatial complexity is defined as the number of groups in the classification, if these groups are of equal size and if they are connected in space. The authors find that one of their own classification schemes (called WEC) performs best on these criteria and recommend it to others. I have read this paper with interest and I think (hydrologically-informed) climate classifications can and should be further developed. I have however various serious concerns about the experimental design the authors present in this manuscript and about the presentation of the work in general. Briefly, I think the authors can do a much better job in explaining how their preferred climate classification scheme was developed. More importantly, I believe that their experimental design makes it a foregone conclusion that the proposed WEC scheme beats the other classification approaches, because the WEC setup uses the same data as are used for WEC evaluation. The manuscript also lacks in clarity. Especially the setup of WEC is difficult to understand. I have outlined these concerns in more detail below.

Major

I here summarize my main concerns about the experimental design. Line-by-line comments are provided below. There is a certain amount of duplication between this section and the line-by-line comments. 1. It is unclear how ETA, ETV and ETC are different in concept The authors propose three mono-variable clustering approaches, based on

[Figure]

ET. To me, these seem like the same approach with only very minor differences in underlying details. I recommended that the authors make clearer why these minor differences are enough to treat these three schemes as completely independent. More details in line-by-line comments below.

Response: As suggested by the reviewer, we have added clarifications on this point. Details are provided in the line-by-line comments below.

2. The WEC description is too brief I struggled to understand exactly how WEC is set up and am left with many questions after reading the Supporting Information to this paper. I strongly encourage the authors to provide more specifics about their methods, both to let the reader better understand the results presented in this paper and to allow the reader to reproduce the WEC results if they wish. More details in line-by-line comments below.

Response: As suggested by the reviewer, we have expanded the explanation in the text and SI (more details line-by-line comments below).

3. Evaluation data is not independent from the data used to set up the four new schemes The authors use data from the CRU TS data set (P and PET) and simulations from the TerraClimate data set (ET) to setup the four new classification schemes. This same data (P, PET and ET) is also used for 3 out of the 5 climate coherence evaluation criteria, where all eight schemes are compared on their ability to group this data into groups with little within-group variability. Unsurprisingly, the four new schemes that are all explicitly conditioned to create such groups with that particular data do very well in this assessment. This comparison is meaningless and should be removed from the manuscript. More details in line-by-line comments below.

Response: Based on the reviewer's important points, this has been added to and addressed in the limitations section of our main text (lines 318-326).

TerraClimate simulations of streamflow (Q) are also used to evaluate the eight classification schemes. These Q simulations are the result of forcing a very simply hydrologic model with P and PET data which, critically, are in part obtained from the same CRU TS dataset that is used to create the four new classification schemes. The TerraClimate ET and Q data can not be seen as independent of the data used to setup the WEC scheme and this undermines the conclusion that WEC is the superior climate classification scheme. If the authors are seeking to thoroughly evaluate the capability of their cluster-based classification, independent evaluation data is needed. This might be obtained from observations of streamflow (instead of model results, the global GSIM database might be of use) or actual evaporation rates generated by modern landsurface schemes. The simple model that underlies the TerraClimate ET and Q values has been first used in 1948 and hardly modified since, and I expect that by now something better may be available. More details in line-by-line comments below.

Response: This is addressed in tandem with the point above. Please see the limitations section of our main text (lines 320-326).

4. The spatial complexity criteria are not well justified and possibly need changing or removal The authors define criteria that reward classification schemes that return large, connected zones with a single internal climate and uniform sizes. However, the main aim of climate classification schemes is to find locations that have similar climates, not necessarily locations that are spatially connected to one another. Neither do I see much reason to assume that all climate groups must be of equal size. These must be better justified or removed. More details in line-by-line comments below.

Response: The reviewer raises an important point. Expanded justification for complexity evaluation is added in section 2.1.

5. KHC conversion to discrete categories is too simplistic The authors convert the continuous climate classification by Knoben et al. (2018) into a discrete climate classification for comparison purposes. Rather than using a clustering approach as is done in Knoben et al. (2018), the authors instead "round pixel values until 30 climate classes

are created". This sounds somewhat simplistic to me and I recommend instead that the authors use some form of clustering to overlay separate climate classes over the continuous hydroclimatic spectrum (KHC). I suspect that this rounding procedure is one of the main causes why KHC shows very high numbers of patches in this study and why Koppen-Geiger (KPG) and KHC appear equally coherent on the Q variable in this study, whereas KHC is substantially better than KPG for finding similar Q regimes according to the assessment in Knoben et al. (2018). More details in line-by-line comments below.

Response: As suggested by the reviewer, we re-created the zones using the gridded inputs (i.e., the three climate indices) provided from Knoben et al., 2018. While patchiness decreased for KHC, it was still statistically significantly higher than the other systems, and CV(Q) remained statistically indistinguishable from KPG.

6. The discussion is somewhat limited I think the discussion would be stronger if the authors address the question "what did we learn about the world?" from their analysis. More details in line-by-line comments below.

Response: As suggested by the reviewer, the end of the discussion has been expanded in the new Conclusions section (lines 329-340).

Line-by-line comments L52. It is equally possible that actual ET rates are hardly used in classification schemes because observations of actual ET are hard to come by at the spatial scales where climate classification is typically useful. Global products of actual ET are typically the result of model simulations or derived from model simulations across this domain. Depending on the origin of the actual ET data, one could argue that a classification that uses this kind of data is more of a modeled climate classification and is critically reliant on the accuracy of the model simulations with respect to actual (meaning occurring in reality) ET.

Response: As suggested, this is updated and addressed. Please see the updated limitations section of our main text (lines 315-327).

L82. The provided reference seems to say that TerraClimate data are at a monthly resolution. If this is the case, how was this disaggregated into daily data?

Response: This was an error attributed to a holdover from earlier preliminary analyses with another source that included disaggregated daily data. This has been corrected. We thank the reviewer.

L89. KPG has not been defined yet.

Response: Corrected accordingly.

L95. Should "monthly mean" be "annual mean"?

Response: This has been clarified (line 116).

L101. Mentioning just the mean R^2 values and their standard deviations seems quite a short description of the accuracy of these fits over 61000+ grid cells on the planet. A somewhat longer description is appropriate. Do the authors' results for P match those of Berghuijs & Woods (2016) in terms of accuracy and spatial patterns of high/low accurate fits? In which regions are the authors' fits more/less accurate?

Response: As suggested by the reviewer, we have added information about p values (the results are highly significant, lines 124-129).

L107. "veteran" seems an odd choice of words. Maybe "legacy"?

Response: We thank the reviewer. This is a better choice.

L111. It is not entirely clear to me how pixel values can be rounded into creating a distinct number of categories. Clustering of the KHC climate index values (similar to how WEC is created) would be much more appropriate and is also the approach taken in Knoben et al. (2018) to create discrete categories that overlay their hydroclimatic continuum. See Figure 3 in Knoben et al. (2018). Can the authors clarify their rounding approach and justify why they use this over a clustering approach that uses the actual data of the KHC scheme?

Response: We have revised our this approach as suggested by the reviewer (see lines 141-145). Instead of rounding values from the provided climate continuum from Knoben et al., 2018, we used their supplied gridded climate indices and employed a k-means clustering approach (number of starts = 80) to form 18 zones, the same number chosen in Knoben et al., 2018. The specific values in our coherence and complexity results were changed somewhat, but the overall narrative of our story did not change. However, we agree that in order to properly build on the proposed system of Knoben et al., 2018, delineating zones should be conducted as outlined in the paper.

L122. I don't quite follow the arguments presented here that should support the idea of a classification that has an equal number of pixels in each zone and I think these arguments need to be clarified or changed. See my concerns with each individual argument below. #1. Koppen-Geiger (K-G) has "relatively high spatial non-uniformity [. . .] resulting in highly variable relevance for regional analysis." I interpret this as meaning that K-G has climate classes of non-uniform size and that this is not useful if one's region of interest falls entirely within a single K-G class out of the ∼30 or so possible K-G classes. I don't quite understand how dividing the entire globe into only 15 classes addresses this problem, as using 15 classes compared to 30 necessarily means lower granularity in the authors' ETA scheme than is possible with K-G. Additionally, I see no reason to assume that forcing each of the 15 ETA classes to have an equal number of pixels will necessarily mean that "regional" analyses have more useful climatic information available. If we consider 5 continents globally, this scheme roughly divides each continent into 3 classes. Does this really help regional studies?

Response: Based on the reviewer's suggestion, the rationale for this has been expanded (lines 155-161).

**2. "Additionally, it is useful to have a simple baseline framework upon which to compare other systems." I agree in principle, but I'd argue that even a simple baseline needs to be somewhat plausible. Dividing a map into 15 regions of equal size, based on a single variable seems a pretty low threshold to beat. Why not use an existing**

classification as a benchmark?

Response: We have referred to the Koppen-Geiger classification system (the existing framework) as our main benchmark. The rationale for using this ETA system as an ET-based benchmark is noted in line 159.

**3. "Zones should ideally be delineated in one piece." This seems somewhat counter-intuitive to me, because the express purpose of climate classification schemes is to identify regions that are similar in terms of variables X, Y and Z. Whether such regions are spatially connected is irrelevant. This also directly c argument #1, where having large areas of a single climate class is mentioned as a negative aspect of the K-G scheme.**

Response: Additional explanation has been updated in lines 157-161.

L124. It is unclear to me how the proposed ETA scheme should be interpreted in a temporal sense. Given that the number of pixels in each class should be the same, this means that the resulting 15 classes are only valid for a given snapshot in time. If the chosen time period for this classification changes, the underlying ET data would change, and wouldn't therefore the number of pixels in each class change too? If we do not want to violate the "equal number of pixels in each class" concept, this means that classes need to be redefined when the underlying time period changes and thus the classes do not have a consistent meaning for different time periods. Imagine a theoretical case where PET uniformly increases over the planet with a constant value. The lowest ETA class now corresponds to a very different real-world climate than it did before.

Response: This has been commented on in lines 322-324.

L124. Using number of pixels is not necessarily a way to guarantee a relatively even distribution of zones in terms of area (which the name "ET Area-optimizing hints at). Based on the CRU data, each pixel represents a certain area on a regular lati-

**HESSD**

tude/longitude grid. Pixels in such a grid do not translate easily into real-world areas. A single 0.5x0.5 degree pixel at the equator might be approximately 50 kmˆ2, while the same pixel size near the poles would represent a fraction of that area. Can the authors clarify why having even distributions in the number of pixels is desirable, even if this could potentially lead to a very uneven distributions of zone size in kmˆ2?

Response: The reviewer brings up a valid point in that it is correct that pixel size changes with latitude. For purposes of this study, pixel distribution captured the overall sentiment of broadly comparing zone sizes.

L133. I don't quite understand what makes ETV different from ETA. ETA imposes groups on the empirical ET CDF. As a result, each ETA group consists of regions with similar ET values (i.e. low CV within the group). ETV seems to minimize within-group CV of ET values by imposing groups on a normal distribution fitted to the empirical CDF. It seems to me that the only difference between ETA and ETV is that ETA uses the simulated ET values (from TerraClimate) directly, whereas ETV uses an approximation of these ET values.

Response: The description of these methods has been expanded (lines 155-161) to clarify that both methods seek to reduce within-zone ET variability, while one focuses only on reducing ET variability (ETV) and the other seeks to evenly distribute pixels across ET variability (ETA).

ETA with 10 groups has CV = 0.2 (Fig S1B); ETV with 10 groups has CV = 0.2 (Fig S2). What does the fitted normal distribution add to this analysis that makes ETV with 29 groups (as determined on line 144) substantially different from ETA with 29 groups?

Response: As suggested by the reviewer above, we have added clarification about the distinction between ETA and ETV in terms of motivations for developing these systems. Regarding number of zones, the data in Figures S1 and S2 (and the corresponding power function regressions) support that CV decreases with additional groups more rapidly for ETV than for ETA (the regression exponent is doubled).

L136. Is "S1" the correct cross reference? I don't see a fitted cont. uniform distribution in Fig S1.

Response: As suggested by the reviewer, this is corrected to show a fitted uniform distribution in Figure S2.

L144. Why did the authors choose to use 29 zones? To me it currently sounds that to maximize within zone ET coherence, one would simply keep imposing more groups until each zone contains a single pixel and within-zone ET CV equals 0.

Response: Lines 179-180 describe that this was the value at which CV decreased the most with the lowest increase in number of zones, as supported by Figure S2.

L145. I'm again a bit confused about the difference between this approach and the preceding ones. Constructing an (empirical) CDF already puts locations with similar mean ET values close to one another, which is also what the clustering in ETC tries to achieve. Additionally, because global mean ET values are approximately continuous (as evident from Fig S1B, S2 and S3), K-means will be trying to impose distinct boundaries on continuous data and therefore tend to gravitate towards clusters of approximately equal size. The authors have already defined ET zones of equal size with minimal within-zone ET variability in their ETA approach. So what does using a clustering algorithm add?

Response: As the reviewer suggests, the value of ETC is now included in lines 188-189.

Equally, comparing Figure S2 and S3 seems to show that ETV and ETC generate approximately the same CV for the same numbers of clusters/groups, but with some scatter in the ETC values (potentially caused by the initial guesses for cluster centroids, see comment below). L146. K-means clustering is quite dependent on the initial guess of cluster centroids for the location of the final clusters. A multi-start framework shows to what extent this vulnerability influences the final clusters. Was the K-means algorithm used in a multi-start framework? If not, why not?

Response: We did use a multi-start framework, which is now noted in line 184.

L150. Is there any particular reason why CV = 0.1 makes a good threshold?

Response: We added in line 189 that we selected this as a sufficiently low CV value.

L151. Should "systems" be "system"?

Response: This is corrected to "system."

L162. I find the description of this new classification scheme in the SI too brief to understand in detail what's going on. I gather that the authors used K-means clustering on various combinations of data but the rest of the method escapes me. This must be addressed, because it (1) makes the authors' claims that WEC is the best out of the 8 classification schemes difficult to assess; and (2) makes the classification impenetrable to others who might wish to reproduce or use this work.

Response: As suggested by the reviewer, we have added clarifications in lines 195-204, lines 206-211, and SI.

Some of the questions I currently have upon reading the SI:

1. How were the combinations of P, PET, ET, Q and delta theta determined? I notice that not all possible combinations are present in Table S1.

Response: As suggested, an explanation for this can now be seen in SI, under the new heading "Multivariate climate classification system selection."

2. What were the K-means settings? Was the algorithm restarted multiple times to test cluster stability?

Response: The reviewer is correct, this was a multi-start framework (n=80 starts), which is now noted in the text (line 184).

3. How were the thresholds for coherence chosen (SI, page 3)?

Response: Explanations for these thresholds are now expanded in SI.

4. What does (1.50) refer to in "KPG(1.50)"? Is this the 50% deviation mentioned in the main text? If so, this should read "KPV value * 1.5". Also, the use of "=" signs is extremely confusing in the lists on this page and should be removed.

Response: The reviewer is correct, this is the 50% deviation mentioned in the main text, and these suggestions have been incorporated in SI.

5. Why does the reader need to know the KPG coherence scores at this point in the analysis?

Response: We have added further explanation in the SI that these are the criteria used to choose the proposed multivariate clustering classification system.

6. Is "number of parameters" equivalent to number of K-means clusters? If not, which K-means parameters are meant?

Response: We have clarified in the main text (line 275) and SI that this means "input variables."

7. If number of parameters is not the same as number of clusters, how was the appropriate number of clusters chosen?

Response: This has been expanded in SI.

8. What does "number of patches" refer to? How was its threshold determined?

Response: This has been expanded in SI.

9. I don't understand how/why the P,PET clustering system was chosen out of all possible options, nor why 22 zones are considered appropriate (SI, page 3).

Response: This has been updated for clarity.

10. The caption of Table S1 states that cells with a "1" in them indicate a combination of variables and number of clusters that meet the criteria specified on page 3 of the SI.

[Figure]

From Table S2 however, it seems that neither ET,PET not P,PET meet the "number of patches" criterion. Why are these then shown in table 1 as if they do meet all criteria?

Response: We thank the reviewer for pointing this out. This inconsistency is an error, a lingering result from prior evaluations (i.e., different methods). This is an important bit that has now been updated.

11. Similarly, neither ET,PET nor P,PET meet the CV(ET) criterion (all values are > 0.33), and all but one fail the CV(Q) criterion (only P,PET with 22 zones has CV(Q) < 1.31). Why does table S1 show these results as meeting the criteria? Why define criteria at all if they are not used?

Response: This has been rectified.

12. I don't understand why there are gaps in Table S1 between certain rows with "1" in them. For example, column P,PET. If 22 clusters are sufficient to meet all criteria (indicated by a "1"), and all criteria are aimed at minimizing differences within a single cluster, it is logically impossible that using 23-27 clusters gives worse results than using 22 clusters, especially considering that 28 and 29 clusters suddenly do meet all criteria again. The only explanation I can think of is that certain settings in the K-means algorithm prevent it from finding the most optimal cluster configuration when 23-27 clusters are used. The multi-start issue I have mentioned before is a possible culprit.

Response: The reviewer brings up a valid point, and we have now added an explanation in the SI for the discontinuities in Table S1, noting that within a clustering framework, the addition of more zones inherently reduces variability, but complexity (i.e., pixel distribution and mean number of patches across zones) is not continuously "better" with the addition of zones.

13. Why were ET,PET and P,ET chosen for further analysis in Table S2 and not others?

Response: We have added this explanation in the main text (lines 206-211).

L158. "Zone complexity" is undefined up until this point and the reader can only guess at what this means by reading the SI. I suggest to clarify what is meant by this in the main text and to also explain whether it is low or high zone complexity that is desirable.

Response: As suggested by the reviewer, this section has been moved and expanded accordingly (section 2.1, lines 82-86). Although we also note that zone complexity was also described in the introduction (lines 71-77).

L160. The SI could use a header to indicate where this section starts.

Response: As suggested, a header has been added.

L164. This section seems to be the justification for many of the authors' methodological choices, in particular about their selection of classification schemes. I suggest to move section 2.6 to the beginning of section 2, so that reader already has access to this information before it is needed to understand the authors' methodology.

Response: This section has been moved as suggested.

L168. The authors argue that "classification systems should consist of a relatively even distribution of pixels across zones, avoiding disproportionally large or small zones" (similar to an earlier argument in section 2.4). I don't understand this argument for two separate reasons: 1. I don't think there is much reason to assume that climatic zones should follow an even distribution across the planet, either in terms of pixels or in terms of area. Globally, deserts are big and alpine regions are relatively small. A classification scheme that tries to create climatic zones of equal (pixel) size will not capture either climate properly, and thus offer little in terms of hydroclimatically relevant information.

Response: This rationale has been expanded (lines 157-161).

2. Like I argued before, I don't know if number of pixels is an appropriate unit here. KPG polar zone ET might take up a fair number of pixels on a regular grid, but in terms of total area the arid classes dominate (compare Sahara size is $\sim$9.2 million km^2,

whereas Greenland is ∼ 2.2 kmˆ2). I don't think pixels are a particularly helpful unit for this analysis. I recommend to remove this criterion from the analysis.

Response: We appreciate the reviewer's thoughtful suggestion. This is a reasonable argument. However, number of pixels is only a general measure of size, and the main objective of reducing size variability was to ensure visibility of smaller zones, not necessarily normalize the dominance of larger zones.

L169. The authors argue that "Zones should be as hydrologically continuous as possible (Meybeck et al., 2013), minimizing patchiness or fragmentation". Like I argued earlier, this is counter intuitive to me. Climate classification systems are intended to find places that are similar climatically, regardless of physical distance. By penalizing systems for patchiness, the authors effectively favor classification schemes that generate large areas of single climate zones, without providing any justification that such schemes are more representative of the real world. In effect, the less data a scheme uses, the more likely it is to generate large connected areas of climate zones and thus, according to this criterion, the better this scheme is. This seems extremely counterintuitive to me and I recommend to remove this criterion and to re-do the analysis. L185. Like before, I'm really not sure why different numbers of pixels contained in different climate classes is a bad thing. If different climate types cover differently sized areas on a map, than that's simply how the global hydroclimate is. Re-drawing the climate class boundaries to create equally large zones is not adding any new hydro-climatic insight to the problem (in fact, one might argue that such an approach uses less hydro-climatic insight).

Response to the previous two comments: The objective of this study is to assess the tradeoffs between complexity and coherence, which has been expanded in the text (lines 82-85). A less complex system is not deemed best if it forfeits hydroclimate coherence. Please see our previous comments to this topic above.

L204. Seeing the authors' assessment scheme indicates some serious methodological

concerns, centering around the fact that no real independent evaluation data has been used. 1. Classification schemes are evaluated on their ability to create low within-zone variability of ET values (i.e. low CV(ET)). This is the exact same data that has been used to condition the authors' ETA and ETC schemes on and, unsurprisingly, when one specifically sets out to create groups with as low as possible within-group ET variation and then uses that same data to see how well that worked, these schemes are impossible to beat. In my opinion the CV(ET) comparison is meaningless because it cannot reasonably be expected that any scheme beats the ETA and ETC schemes in this comparison.

Response: We thank the reviewer for this suggestion, which is valid, to an extent. While it may be expected that CV(ET) is lowest for ET-based schemes, it is important to include this metric is for relative comparison purposes. We are addressing the question of how much better are the systems in which ET is directly optimized compared to other previously established systems? There is valuable information to be gleaned from that.

2. The same argument can be applied to the CV(P) and CV(PET) criteria. The authors' WEC scheme is specifically conditioned on creating groups with low variability in these two climate indicators and therefore the comparison with existing classification schemes is meaningless. The only question this seems to answer is "are established climate classification schemes better at clustering P and PET values then a clustering algorithm can cluster P and PET values?" Apparently and not entirely unexpectedly, they are not.

Response: Similar to a response above, this was more to relatively compare "how much" better is WEC than other systems not directly using mean annual P and PET as input variables.

3. The ET and Q data in this study are taken from the TerraClimate dataset (Abatzoglou et al., 2018). To quote directly from Abatzoglou et al. (2018): "A one-dimensional modified Thornthwaite-Mather climatic water-balance model (WBM)22,31 was used to

calculate monthly water balance from 1958–2015. The WBM is a single bucket model applied consistently across global land surfaces that operates on a monthly time step and considers the interplay between precipitation, ET0, as well as soil and snowpack water storage. The WBM accounting scheme considers runoff as the excess of liquid water supply (precipitation and snowmelt) used by monthly ET0 and soil moisture recharge. Soil water is extracted during months where ET0 exceeds liquid water supply, with the extraction efficiency of soil water declining exponentially with the ratio of soil water to extractable soil water capacity. Under such conditions, actual evapotranspiration is counted as the liquid water supply plus the soil water utilized and climatic water deficit is the difference between ET0 and actual evapotranspiration." Due to its simple design and monthly time step, this model has little to no capacity to generate non-linear (and thus realistic) hydrologic behavior. It is thus likely that these simulated Q and ET data are strongly correlated to the forcing data used to generate them. TerraClimate uses the CRU TS v4.0 as one of its inputs, while CRU TS v4.0.4 is used by the authors to provide the P and PET data for their classification. WEC is thus conditioned on a dataset that is very similar to the dataset used to generate the ET and Q data that are used to evaluate the different classification schemes in this paper. It is therefore not entirely surprising to see WEC perform (reasonably) well on the CV(ET) and CV(Q) criteria. In a comparison such as this, independence of the evaluation data is critical to guarantee a fair comparison. I suggest to replace the evaluation data for Q with observations from for example the global streamflow attributes dataset GSIM, or those from the Global Runoff Data Centre. ET might be obtained from modern land surface models run at a global scale, instead of from a bucket model first conceived of in 1948 and hardly changed since.

Response: Similar to a response above, while the runoff data is generally just long-term P minus ET, TerraClimate ET has been validated at sites across the globe. This is feasible, since we are investigating long time scales and large spatial scales, where interannual and smaller spatial variability is more muted. Broadly, this data is sufficient for our analysis (see lines 319-327).

4. As argued before, neither having equal numbers of pixels in each zone nor having a low number of patches seem particularly indicative of a useful climate classification scheme to me. I suggest to remove these if they cannot be better justified.

Response: Please see our previous responses pertaining to our logic aligned with that of Meybeck et al., 2013 (section 2.1, lines 82-93 lines 156-164 in the main text).

5. The fact that ETA has "the fewest number of zones and very high coherence for zone size (by far the lowest CVz)" (L200) should not come as a surprise when the entire premise that underlies ETA is dividing the available number of pixels in a small number of zones of equal sizes.

Response: Please see previous response to this comment.

6. It is odd to see that ETA has the best score for "zones" (indicated in bold), because it has the lowest number of zones. This is counterintuitive to me, because this seems to favor climate classification schemes that are explicitly not good at their intended purpose, namely to define different climatic regions. The ideal score on this assessment would be obtained by a scheme that returns 1 climate zone, which means that all climates are classified as being the same. I recommend to remove this criterion.

Response: We thank the reviewer for analytically and critically examining this aspect of the analysis. Within the context of "trade-offs" (lines 83, 170, and 201) the number of zones should be considered alongside their ability to minimize within-zone hydroclimate variability. A classification system in which only one zone is delineated would indeed have low complexity, but it would not yield high coherence. Conversely, a system with an infinite number of allowable zones would yield highest possible coherence, but this would be meaningless and therefore inefficient (just as one zone would be meaningless and inefficient). Our objective here was to find the equilibrium between number of zones and hydroclimate coherence. As shown in our SI figures, we selected the lowest number of zones with highest level of coherence.

Interactive
comment

7. Concluding, I believe that the evaluation criteria heavily favor the authors' classification schemes and that the fact that WEC performs well on them is no more than an artifact of the experimental design. The evaluation data cannot be considered independent from the data used to generate WEC (and ETA, ETV, and ETC) and the pixel, patches and zones criteria are not well-justified. This undermines the authors' conclusion that WEC is a better classification scheme than Koppen-Geiger or any of the other existing classification approaches. Independent evaluation data is needed.

Response: Independent validation data would add more to this narrative, but within the structure of our story, we argue that it is not entirely necessary. Please see our previous responses above (lines 319-324 in main text).

L205. The caption mentions that certain schemes have statistically similar results. How was this determined?

Response: The caption states that K-S tests were applied to determine statistical difference across CV distributions.

L205. I find the number of patches for KHC extremely high. Can this be a consequence of the "pixel rounding" procedure described on line 111? This result indicates that this rounding procedure is unnecessarily crude, because the KHC groups shown in Knoben et al. (2018) are not nearly as fractured as this. Given the authors' use of actual clustering algorithms in this paper, applying those to the KHC data seems feasible.

Response: This has been rectified. As the reviewer rightly suggests, KHC boundaries were redrawn based on the gridded climate indices provided by Knoben et al., 2018 and the k-means clustering approach. Mean number of patches therefore reduces from 275 to 97.

L214. This grouping procedure needs to be explained in more detail than is currently done. Is this the result of another k-means clustering exercise? Is phi_bar the mean aridity over the group?

Response: We have clarified the text (line 247) that phi_overbar is mean aridity of zones within each group.

L216. Where are these relationships shown? How do these compare to those from Koppen-Geiger classes?

Response: This has been removed.

L217. "... indicating higher spatial variability (lower coherence) with increasing aridity." I'm not entirely sure how to interpret this statement. Which conclusion should the reader draw here?

Response: This has been removed.

L226. This discussion seems to imply that WEC provides higher granularity in certain areas. I think this argument can be reversed too: whereas KPG shows some longitudinal diversity in climate zones, WEC lumps London, Amsterdam, Berlin, Warsaw, Moscow and a substantial part of the Russian Federation into a single climate zone. I think a more balanced discussion of WEC is needed.

Response: As suggested by the reviewer, this section has been updated to reflect better geographic balance (lines 261-263).

L235. "Discussion and conclusions"

Response: This has been corrected to include a separate "conclusions" section.

L238. This statement is a bit inaccurate, because Knoben et al. (2018) use the monthly ratio between P and PET as one of their predictors.

Response: This clarification is now reflected (line 274).

L243. It needs to be clarified what is meant by "parameters" here. This seems to say that the parameters used by WEC are mean P and mean ET. What about the number of clusters used and other k-means settings?

Response: This has been clarified to mean number of input variables (line 275).

L243. "a notable aspect of system complexity" This argument is unclear. Why is the number of parameters relevant in a climate classification scheme? With the definition of "parameters" as "number of climatic variables used" (as I gather from this section), this sentence seems to imply that the less data used, the better the classification scheme is. Can this be clarified?

Response: This has been changed to reflect input variables (line 275).

L245. It should be no surprise that WEC, as the result of a k-means clustering algorithm applied to P and PET data, consists of groups that have low internal variability on that same P and PET data. Also, Table 1 seems to say that KPG scores better on CV(ET) and CV($\Delta\theta$) while WEC scores higher on CV(Q) only, albeit with a higher standard deviation than KPG gets.

Response: Table 1 shows results based on a K-S test (noted in caption), in which the entire distribution of each metric is evaluated. Therefore, WEC is either as good as or better than KPG in all coherence and complexity metrics (lines 234-236).

L250. I find this odd. As the authors say (lines 48-49), Knoben et al. (2018) compare their classification scheme to KPG and find that KHC is better for hydrologic purposes (in this case for grouping locations with similar hydrologic regimes expressed through a variety of metrics). In the authors' words, KHC is "more coherent with respect to Q" then KPG. Why does this not show in the authors' assessment? This could be related to "pixel-rounding" or to the use of simulated Q in this study.

Response: As suggested by the reviewer, and as noted above, we have revised the methodology for KHC zone formation. After reforming the boundaries based on k-means clustering and using the provided climate indices from Knoben et al., 2018, Q is in fact found to be more coherent, but it is still not statistically different from KPG. Again, we need to consider that the present analysis is based on long term mean annual rates

and does not account for Q variability/regimes like Knoben et al., 2018.

L250. Additionally, if KHC is not high in Q coherence, why is it indicated as one of the three best systems in Table 2?

Response: We have revised this to improve clarity (line 284-288).

L252. I suggest to remove these statements because they are trivial. If one clusters ET data one gets groups with high ET coherence. If one divides all data into 15 equal-sized groups, one gets groups with equal sizes. This should not need to be presented as a benefit of these approaches.

Response: We added a qualifier to reiterate that indeed these results are in line with the objectives of each ET zoning framework (lines 290-291).

L257. Which variables do the authors mean when they refer to "water budget dynamics"?

Response: This indicates all variables, so this has been changed from "water budget" to "hydroclimate" (line 295).

L265. I don't quite follow this sentence, perhaps because I don't understand which variables are meant with "water budget components".

Response: This has been clarified in the text (lines 301-302).

L283. Given all the methodological concerns that need to be addressed, I do not think this conclusion is currently well supported.

Response: We have revised the methods in accordance with the reviewer's suggestions (i.e., KHC zone formation).

L289. This seems overly optimistic to me. Which kind of environmental management decisions can be made at a scale where a handful of Koppen-Geiger classes or WEC groups provide a useful indication of local conditions?

Response: This sentiment has been clarified in lines 335-340.

L289. What I miss in this discussion is a critical assessment of hydroclimatic understanding and how the authors' proposed WEC adds to this. Existing classification schemes are based on hypotheses about how the world works and about which elements of the global climate are first-order controls on the resulting hydroclimate. WEC is simply a clustering method that finds regions with similar P and ET values. What does this teach us about global hydroclimatic relationships? If WEC is better than established methods that do rely on theory, then where is this theory faulty or incomplete? In other words, I think this discussion would be much stronger if the authors where to consider the question "what did we learn about the world?"

Response: As suggested by the reviewer, these revisions have been included in section 5.

---

## Author Comment (AC3) · 31 Dec 2020

We thank the reviewer for these helpful comments. The Reviewer comments are listed below, along with our response to each. Most comments require only minor edits. In some cases, we describe revisions to the manuscript (with line numbers), and we recognize that the revised manuscript is requested in a subsequent step.

Reviewer 3

The authors highlight the absence of a proper ET-derived classification system. It is a such that they propose a "series of ET-based global classifications that should yield

comparatively higher ET coherence than other systems", by assessing coherence and shape complexity within the classifications. I find that the study is well carried out, very nicely written and illustrated. Methods are also well explained. I think that the study would be a great contribution to HESS, enabling researchers to choose the best system of classification suited to their purposes, specially regarding ET. However, I have some clarifications that need to be added to the former version of the manuscript.

Response: We thank the reviewer for the positive comments.

1. Existing so many proxies for landscape connectivity assessments the authors need to support the use of "zone area" and "zone fragmentation". Why are these the best ones? Also, I did not see any formula, or explanation. This is needed wince these classification schemes are not that well known in the field of water resources, and rather in the field of ecology. A figure would be useful.

Response: This has been expanded in section 2.1 (lines 80-97).

2. ET is dependent on many local parameters related to land cover and land use (See for example Sterling et al., 2013). Human activities such as agriculture, urbanization, deforestation, heavily affect these parameters and then would imprint less coherence, more variability and patchiness into the classification system. The authors should comment/adjust on this.

Response: We now comment on this (lines 294-296). The suggested reference has been incorporated.

3. Furthermore, I would have done the analysis with a more "large-scale climatic parameter" that involves less spatial variability (and more spatial coherence-less CV) at the local scale, such as the aridity index (PET/P) or evaporative ratio (ET/P). Have the authors considered this?

Response: As the reviewer suggests, forming zones based on the aridity or evaporative index can be useful. As we use mean annual P, PET, and ET as metrics of coherence,

aridity and evaporative indices are thus captured indirectly. Note that in preliminary analyses we used aridity index to create zones but we found this to be a less robust way of forming zone boundaries compared to mean annual P and PET separately.

4. I was expecting a more concrete recommendation on the best system for ET. Which one is it if you have to choose one?

Response: As suggested by the reviewer, WEC is now more clearly put forward as our recommended framework (section 5, lines 333-340).

5. Conclusions are missing, and should be independent from the Discussion.

Response: This has been corrected.

References: Sterling, S.M., Ducharne, A., Polcher, J., 2013. The impact of global land-cover change on the terrestrial water cycle. Nature Climate Change 3, 385–390. https://doi.org/10.1038/nclimate1690

---

## Author Response (AR2)

**Manuscript title: Coherence of Global Hydroclimate Classification Systems**

**Manuscript number: hess-2020-522**

**Reviewer 2**

I find the provided responses to review comments insufficient in several places. I had outlined six major comments in my previous review. Below are brief descriptions of which ones I think deserve further consideration.

---- The lack of independent evaluation data is a critical flaw of this study.

The discussion now includes several sentences that mention that the proposed new classification systems are evaluated on data that is either the same or dependent on the calibration data, which in my opinion is insufficient to address this concern. I consider the lack of independent evaluation data not a limitation but a methodological flaw that needs to be addressed before this paper can be published. Without independent evaluation data, no honest comparison between the established and new schemes is possible.

> Response: We thank the reviewer for the emphasis to incorporate independent validation data. Please see lines 111-119 for these updates, also shown below:
>
> "Additional ET and Q datasets were used for independent validation purposes. Observation-based monthly Q from 1980-2014 were obtained at 0.5° x 0.5° resolution from monthly global gridded runoff data (GRUN, Ghiggi et al., 2019). The Global Lobal Evaporation Amsterdam Model (GLEAM) produced terrestrial daily ET for 1980-2020 at 0.25° x 0.25° resolution, which was also resampled to 0.5° x 0.5° resolution. Here, we used the updated GLEAM version 3.5a, which is based on ERA5 net radiation (satellite) and air temperature (reanalysis) datasets, downloadable at a monthly timestep (Martens et al., 2017). The GLEAM ET and GRUN Q datasets were independent from TerraClimate ET and Q datasets both temporally (Figures S1 and S3) and spatially (Figures S2 and S4). The two ET datasets were more similar than the two Q datasets, based on monthly linear models ($R^2$ ranging from 0.78 to 0.87 for ET and 0.47 and 0.84 for Q), and both ET and Q datasets showed spatially consistent seasonal differences."
>
> Results from these data can be seen in Figures 2, S1-S4, and S11.

---- The complexity criteria are still insufficiently supported and clear:

1a. Hydroclimatic zones do not have equal sizes in reality. The justification that zones must have equal sizes for visibility of the small zones in my opinion simply means that zones get redefined to be less hydroclimatically distinct. In other words, regions that could be distinct hydroclimates get merged to conform to the minimum zone size criterion, meaning we lose hydroclimatic insight as a consequence of the zone size requirement. As far as I can tell, this argument is not addressed in either the response or the updated manuscript.

> Response: We agree with the reviewer that the purpose of delineating hydroclimate boundaries is to identify distinct regions that share hydroclimatic similarities, which is not necessarily size-dependent. The justification for ensuring that zone sizes are not meaninglessly small is rooted in disproportionality (line 95). For example, two of the KPG zones disappeared when resampled to

0.5° x 0.5° resolution (line 158), which results in highly variable spatial relevance of zones within the context of regional analyses (line 172).

1b. In addition, pixels are an inappropriate unit of measurement for area, because the area a single pixel represents is not constant within the regular lat/lon projection used in this manuscript. This also means that any pixel-based criterion is conditional on the geographical projection used for the source data and thus not on the data itself. This is avoidable by using actual area values.

Response: We agree that zone area is a much better metric, and our methodology has been rectified to include an assessment of zone areas instead of number of pixels per zone. The results still showed high spatial variability for KPG compared to the novel frameworks (Figure 4B).

2. The authors rely on Meybeck et al. (2013) for the statement that zones should be delineated in one piece. To my knowledge the authors address neither in their replies, nor in the main manuscript my concern that this criterion rewards schemes that are not very good at what they are supposed to do, namely define regions with similar hydroclimates regardless of spatial proximity. Meybeck et al. (2013) have clear reasons for wanting to do so (mostly to not separate a river's headwaters and its lowlands) and an experimental design that supports their reasoning (i.e. using basin shapes in addition to climatic data) which do not seem to apply in this paper. Given that the streamflow data used in this manuscript is simply local P – EP and does not consider catchment aggregation at all, I would say that Meybeck cannot be used to support the use of this criterion.

Response: Meybeck et al., 2013 state that "ideally" zones would be "delineated in one piece," though they recognize this is not physically possible. A clause to reflect this limitation was added to line 177, shown below:

"This type of spatial condition is similar to the prioritizations of the MHR framework that state zones should ideally be 'delineated in one piece,' although this is not a physical reality (Meybeck et al., 2013)."

3. I now understand that the authors use the number of zones as part of a trade-off (given equal hydroclimatic coherence, the system with fewer zones is preferable) but I think this is insufficiently clear in the main manuscript. Most importantly, this tradeoff is neither mentioned in the description of the results (for example Table 1 simply lists ETA as having the best – lowest – number of zones despite it not having very high coherence), nor in the discussion.

Response: We agree that this tradeoff assessment was not made abundantly clear. The methodology has been updated to reflect a more systematic determination of what is considered "good" with respect to number of zones. Please see section 2.6 and Figure 2, where a systematic approach for choosing the optimal number of zones is outlined. Information previously gleaned from the table aforementioned by the reviewer has been transformed into Figures 3 and 4 for enhanced clarity.

---- The introduction and discussion are somewhat limited.

Given the focus on annual or longer time scales, the Budyko framework probably needs to be part of the introduction. Although not a climate classification system in the traditional sense, it is a well-established

way of organizing locations based on long-term aridity (P / EP) and long-term ET and Q. A discussion of the Budyko framework can replace the current sentence on line 68 ("given the major gap regarding the inclusion of ET in climate classification systems, ..." ) and may provide a justification for assessing the existing climate classification schemes at annual or longer scales. This could introduce a research question along the lines of "do existing hydroclimate classification schemes correspond with catchment organization as predicted by the Budyko curve?" or something similar.

> Response: A statement regarding the Budyko framework was added to the Discussion in lines 334-337, shown below:
>
> "To delineate the landscape based on ET dynamics, the Budyko framework is a longstanding, well-vetted mechanism for estimating the evaporative index (ET/P) using the primary drivers of the water budget, PET and P, as represented by the aridity index (Budyko, 1974; Milly, 1994; Reaver et al., 2020a; Reaver et al., 2020b; Zhang et al., 2004)."

I will also repeat one comment directly from the first review: What I miss in this discussion is a critical assessment of hydroclimatic understanding and how the authors' proposed WEC adds to this. Existing classification schemes are based on hypotheses about how the world works and about which elements of the global climate are first-order controls on the resulting hydroclimate. WEC is simply a clustering method that finds regions with similar P and ET values. What does this teach us about global hydroclimatic relationships? If WEC is better than established methods that do rely on theory, then where is this theory faulty or incomplete? In other words, I think this discussion would be much stronger if the authors where to consider the question "what did we learn about the world?" I don't see where this is addressed in section 5.

> Response: The reviewer highlights important points regarding impact. Please see lines 363-377, shown below:
>
> "It is widely accepted that water and energy, chiefly in the form of precipitation and solar radiation, govern long term socioecological water availability at large spatiotemporal scales (Budyko, 1974; Berghuijs and Woods, 2016; Knoben et al., 2018; Sanford and Selnick, 2013). Several previous climate classification systems aimed to represent this water-energy interaction within bounded zones that encompass similar hydroclimatic sensitivities (Knoben et al., 2018; Meybeck et al., 2013). It was concluded here that WEC, using water and energy in the form of P and PET rates, was the best overall system for building zones that encompass similar Q rates. This suggests that the WEC scheme is valuable for assessing and predicting water availability changes given changes in water and energy. Therefore, WEC is the most relevant system for direct management understanding and application as it relates to hydroclimate dynamics.
>
> This work is a promising pathway to regionalization within many different biophysical and socioeconomic contexts, clustering drivers to form zones of similar response variable sensitivities in order to more accurately extrapolate locally derived results and regional impacts of local management practices. The WEC framework can thus inform regional to national scale management strategies in the effort to account for potential hydroclimate zone-dependent responses to climate and land cover changes."